# FlowRefiner: A Robust Traffic Classification Framework against Label Noise

**Mingwei Zhan**[1], **Ruijie Zhao**[2]*, **Xianwen Deng**[1], **Zhi Xue**[1]†, **Qi Li**[3],
**Zhuotao Liu**[3], **Guang Cheng**[2], **Ke Xu**[3]†

[1]Shanghai Jiao Tong University    [2]Southeast University    [3]Tsinghua University

{mw.zhan,2594306528,zxue}@sjtu.edu.cn    {ruijiezhao,chengguang}@seu.edu.cn
{qli01,zhuotaoliu,xuke}@tsinghua.edu.cn

## Abstract

Network traffic classification is essential for network management and security. In recent years, deep learning (DL) algorithms have emerged as essential tools for classifying complex traffic. However, they rely heavily on high-quality labeled training data. In practice, traffic data is often noisy due to human error or inaccurate automated labeling, which could render classification unreliable and lead to severe consequences. Although some studies have alleviated the label noise issue in specific scenarios, they are difficult to generalize to general traffic classification tasks due to the inherent semantic complexity of traffic data. In this paper, we propose FLOWREFINER, a robust and general traffic classification framework against label noise. FLOWREFINER consists of three core components: a traffic semantics-driven noise detector, a confidence-guided label correction mechanism, and a cross-granularity robust classifier. First, the noise detector utilizes traffic semantics extracted from a pre-trained encoder to identify mislabeled flows. Next, the confidence-guided label correction module fine-tunes a label predictor to correct noisy labels and construct refined flows. Finally, the cross-granularity robust classifier learns generalized patterns of both flow-level and packet-level, improving classification robustness against noisy labels. We evaluate our method on four traffic datasets with various classification scenarios across varying noise ratios. Experimental results demonstrate that FLOWREFINER mitigates the impact of label noise and consistently outperforms state-of-the-art baselines by a large margin. The code is available at `https://github.com/NSSL-SJTU/FlowRefiner`.

## 1 Introduction

Network traffic classification is a fundamental task for the management, security, and optimization of modern networks. It enables critical capabilities such as identifying malicious behaviors (1; 2; 3), enforcing quality-of-service (QoS) policies (4; 5; 6), and monitoring application usage (7). It can also support user-centric analysis such as profiling in social network applications by fine-grained inference of user actions (8; 9). As network applications and protocols continue to evolve, traffic data have become increasingly complex, reflecting not only protocol interactions but also diverse user behaviors and service patterns (10). These complexities make automated traffic classification both more important and more challenging (11; 12; 13).

In recent years, deep learning (DL) algorithms have emerged as powerful tools for traffic analysis for handling such complexity (14; 15; 16; 17; 18). By leveraging large-scale labeled raw traffic data, DL-based models can automatically extract discriminative representations and achieve accurate classification performance. However, the success of DL-based methods heavily relies on the quality

---

*Corresponding author.    †Project advisors.

39th Conference on Neural Information Processing Systems (NeurIPS 2025).

of labeled training data, and their performance can degrade significantly in the presence of label noise (19). On the other hand, label noise, caused by incorrect, ambiguous, or inconsistent annotations, is especially a critical issue in traffic analysis tasks. Erroneous labels can hinder generalization in real-world scenarios and lead to severe consequences, such as failing to detect malicious traffic as suspicious or misclassifying latency-sensitive applications in QoS enforcement. Unfortunately, label noise is especially common in realistic traffic due to the complexity and variability of real-world environments and the flaws of automated labeling techniques (20; 21). Though labels of lab-generated traffic could be refined through controlled environments and manual annotation processes, the label quality of realistic traffic is still challenging since the automated labeling is prone to errors (22).

Despite its significance, the issue of label quality is rarely noticed and discussed in existing DL-based traffic classification methods. Only a few studies (23; 24) explicitly target label noise in the context of malicious traffic, a specific subfield of traffic analysis focused primarily on intrusion detection. However, these approaches can hardly generalize to general traffic classification tasks, as some (23) are limited to binary classification, while others (24) rely on strong benign–malicious separability that does not hold in more diverse traffic scenarios. In addition, though various methods (25; 26; 27; 28) have been developed to address label noise in other fields (e.g., computer vision), they still struggle in traffic classification due to the inherent structural complexity and obfuscated semantics of traffic data. Thus, dealing with label noise in general traffic data remains a crucial and unresolved issue.

In this paper, we propose FLOWREFINER, a robust and general traffic classification framework against label noise. Unlike existing approaches from computer vision or malicious traffic detection, FLOWREFINER is tailored to the unique characteristics of traffic data and supports general classification across diverse scenarios. FLOWREFINER effectively detects, corrects, and classifies traffic with noisy labels in a uniform framework, achieving high performance without relying on high-quality labeled data, which are costly and difficult to obtain in increasingly complex network environments. Moreover, by accurately identifying mislabeled flows, FLOWREFINER provides valuable feedback to assist network administrators and annotators in improving traffic data quality. Specifically, FLOWREFINER consists of three key components: a traffic semantics-driven noise detector, a confidence-guided label correction mechanism, and a cross-granularity robust classifier. First, we deploy a pre-trained traffic encoder to extract latent traffic semantic representations for noisy flow isolation. Then, the majority labels within each cluster guide the division of flows into clean flows and noisy flows. Next, the confidence-guided label correction refines the noisy flows by fine-tuning a traffic label predictor on clean data and correcting mislabeled samples based on predicted confidence scores, thus reintroducing the corrected noisy flows into the clean flows to form the refined flows. Finally, our cross-granularity robust classifier integrates both flow-level and packet-level classification, enabling the model to capture generalized patterns rather than focusing on noise-driven isolated features. It ensures robustness and improves classification performance even in the presence of noisy labels. In summary, our contributions are as follows:

- We introduce FLOWREFINER, a robust method for general traffic classification that effectively detects, corrects, and classifies traffic with noisy labels. To the best of our knowledge, this is the first work to address the label noise issues across diverse traffic classification tasks.

- We propose a traffic semantics-driven noise detector. It leverages the traffic semantics extracted by the pre-trained traffic encoder to detect outlier labels as noise, where the raw flows are divided into clean flows and noisy flows.

- We fine-tune a traffic label predictor based on isolated clean flows to perform confidence-guided label correction of detected noisy flows. The correction results can assist network administrators and annotators to improve traffic label quality.

- We design a cross-granularity robust classifier that integrates both flow-level and packet-level classification tasks to improve robustness, preventing the encoder from overfitting to noisy labels.

- We evaluate FLOWREFINER on four real-world traffic datasets with different noise ratios. Results show that our method can achieve robust traffic classification against label noise and significantly outperforms the state-of-the-art baselines.

## 2   Related Work

**Traffic Classification Methods.**    In traffic analysis, traditional rule-based methods initially relied on fixed attributes like port numbers and protocols to classify traffic. However, with the rise of encryption

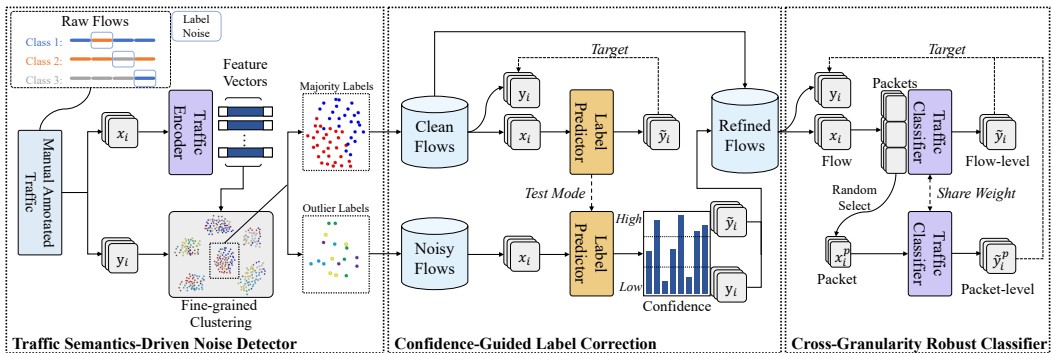

Figure 1: The overview of FLOWREFINER.

protocols such as SSL and TLS, their effectiveness has gradually decreased because these protocols conceal key traffic characteristics (29; 7). To overcome these limitations, machine learning (ML) techniques, including support vector machine (SVM) and random forest (RF), leveraged statistical features from traffic data to automate feature selection and improve classification accuracy (30; 7). Although these ML-based methods marked a significant advancement, they heavily depend on handcrafted features, which limits adaptability across various traffic types (13).

Deep learning (DL) methods have brought a shift towards automatic feature extraction from raw traffic data, with convolutional neural networks (CNNs), recurrent neural networks (RNNs), and Transformers being widely adopted for learning hierarchical representations (16; 14). Recent arts have also explored pre-training techniques, with PERT and ET-BERT (31; 32) applying NLP-inspired BERT models (33) to encrypted traffic classification, and Flow-MAE (34), YaTC (35) using masked autoencoders (MAE) (36) to improve robustness in feature extraction under encryption. However, they are particularly prone to label noise, as their high capacity allows them to overfit incorrect labels (37; 19). Some works (24; 38; 39; 40) have noticed the label noise issue in deep learning-based intrusion detection systems, and propose corresponding methods for malicious traffic. However, they lack generalizability to more general traffic classification tasks. It is urgent to propose an effective method to resist noisy labels for general traffic analysis tasks.

**Label Noise Learning Methods.** Various methods have been developed to address label noise in other fields such as computer vision. A series of approaches like Generalized Cross-Entropy (GCE) (27) and Symmetric Cross-Entropy (SCE) (28) modify traditional loss functions to better handle noisy labels. GCE combines cross-entropy with mean absolute error to make learning more noise-resistant, while SCE balances penalization of misclassified samples. In addition, some works (26) (25) investigate filtering noisy samples that rely on the loss value distribution. On the other hand, data augmentation technique has been wild used against label noise. As a classic robust augmentation, Mixup (41) applies interpolation between samples to generate robust training data. But it struggles when applied to the structured nature of encrypted traffic flows, where such interpolations may not produce meaningful results. Recent methods like Manifold DivideMix (42), which incorporate contrastive learning, offer improved robustness to severe noise. However, they rely on well-defined meaningful data augmentation, which is hardly performed in structured traffic data, limiting their ability to handle label noise within traffic flows.

In summary, while these label noise learning methods offer powerful mechanisms in other fields, the traffic classification still lacks effective solutions that can robustly handle noisy labels. To fill this gap, we propose FLOWREFINER, a robust framework specifically designed to detect, correct, and classify traffic with noisy labels, offering a comprehensive solution for label noise in general traffic classification.

# 3 FLOWREFINER

We introduce FLOWREFINER, a robust method for general traffic classification that effectively detects, corrects, and classifies traffic with noisy labels. Our framework starts with traffic semantics extracted by the advanced pre-trained encoder could reflect the similarities and diversity of flow characteristics,

providing a basis for detecting label-noisy flows (detailed in Sec. 3.1). After that, a predictor is fine-tuned via detected clean flows to correct noisy labels and construct refined flows with low-level label noise(detailed in Sec. 3.2). Finally, a traffic classifier with a cross-granularity structure is designed to perform robust general traffic classification against label noise (detailed in Sec. 3.3).

## 3.1 Traffic Semantics-Driven Noise Detector

Traffic semantics-driven noise detector aims to identify and isolate noisy labels in traffic data effectively. It achieves effective noisy flow detection by leveraging the traffic semantics extracted from the flows and combining their similarity with label distribution.

**Traffic Semantics Extraction.** We first build a traffic semantics-oriented encoder via the pre-training paradigm to extract the latent representation of traffic data. Benefiting from the learned semantics from the unlabeled traffic data in pre-training, the pre-trained traffic encoder can effectively extract traffic semantics (32), while remaining immune to interference from potential noisy labels.

We adopt the Masked Autoencoder (MAE) (36) style pre-training paradigm to enable the encoder to capture the traffic semantics by randomly masking input and reconstructing the missing portions (which is further detailed in Appendix A). In addition, the encoder deploys the Transformer architecture as the backbone for feature extraction. Formally, let $X = \{x_1, x_2, \ldots, x_N\}$ represent the dataset, where $x_i$ is an individual flow sample. The pre-trained encoder, denoted as $f_{\text{enc}}(\cdot)$, processes each sample $x_i$ and produces a $d$-dimensional feature vector: $z_i = f_{\text{enc}}(x_i), \quad z_i \in \mathbb{R}^d$.

**Noise Detection based on Traffic Semantics.**
The distances between traffic semantics embeddings can effectively reflect the similarities in flow content and function. These similarities have the potential to highlight instances of label noise, as similar flows with discrete labels are likely candidates for noisy labels. However, as we visualized via t-SNE in Figure 2, since the traffic semantics have not yet been aligned with the correct labels, there are two main concerns that should be considered during noise detection: There are two main challenges in detecting label

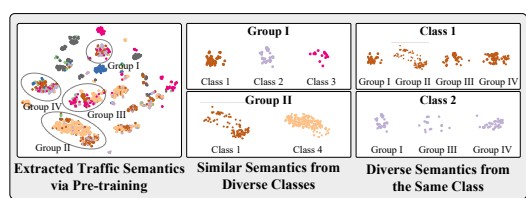

Figure 2: The t-SNE visualization of the traffic semantics.

noise via similarities: (1) Traffic data in the same class may not exhibit consistent behaviors, forming diverse compact clusters in the latent space; (2) Traffic data from different classes may appear semantically similar, leading to groups of mixed-category flows. Therefore, we designed a fine-grained clustering and defined majority labels for each cluster to address the above issues and detect noisy labels.

To capture the different behaviors inside the same class, we further cluster the semantics similarity between flows with a more fine-grained level than categories. Specifically, we partition extracted flow samples into $K$ clusters, according to their extracted feature vectors $\{z_1, z_2, \ldots, z_N\}$ by the K-means algorithm, where each flow sample is grouped based on its semantic proximity to other samples. Especially, the number of clusters is defined as $K = n \times C$, where $C$ represents the total number of classes and $n$ is an integer hyperparameter introduced to control the granularity of the clustering. The objective function for K-means clustering is to minimize the sum of squared distances between the feature vectors and the corresponding cluster centroids:

$$\min_{\mu_k} \sum_{i=1}^{N} \sum_{k=1}^{K} \mathbb{1}(z_i \in C_k) \|z_i - \mu_k\|^2, \tag{1}$$

where $\mu_k$ is the centroid of the $k$-th cluster, and $\mathbb{1}(z_i \in C_k)$ is an indicator function that assigns $z_i$ to the nearest cluster $C_k$. Subsequently, flows within the same cluster share the most similar semantics, and the presence of inconsistent labels within each cluster is regarded as potential noise.

Considering the potential similar activity of flows from different categories, we define the majority labels for each cluster to represent the inside overlapped true classes. Let $L_k$ be the set of labels for the samples in cluster $C_k$. The majority label $L_k^{\text{major}}$ is determined as the top $m$ most frequent labels

in the cluster. Samples with labels that are not among the majority labels $L_k^{\mathrm{major}}$ in their cluster are treated as statistical outliers, and their labels are considered noisy.

Finally, we detect and divide the raw flows into a clean flow set and a noisy flow set:

$$\mathcal{D}_{\mathrm{clean}} = \{(x_i, y_i) \mid x_i \in C_k, y_i \in L_k^{\mathrm{major}}\}, \mathcal{D}_{\mathrm{noisy}} = \{(x_i, y_i) \mid x_i \in C_k, y_i \notin L_k^{\mathrm{major}}\}, \quad (2)$$

where $y_i$ is the raw label for sample $x_i$, and $L_k^{\mathrm{major}}$ is the majotity label of cluster $C_k$. The clean flows $\mathcal{D}_{\mathrm{clean}}$ can be used to support further training, while the noisy flows $\mathcal{D}_{\mathrm{noisy}}$ can be flagged and reported to the manager or annotator for review.

## 3.2 Confidence-Guided Label Correction

Through the semantic-driven noise detector, most noisy flows are identified and isolated from the clean flows according to their label-independent traffic semantics. Furthermore, this clean flow set can provide valid supervision to further correct the label distribution of the noisy flows and expand the available training data. For this purpose, we design the confidence-guided label correction module to correct flow labels in the noisy flow set based on prediction confidence.

**Traffic Label Predictor Fine-tuning.** In this stage, we aim to fine-tune the latent representations of flows to align with the labels of selected clean flows $\mathcal{D}_{\mathrm{clean}}$, thereby obtaining a confidence-aware traffic label predictor that can assess the category of selected noisy flows $\mathcal{D}_{\mathrm{noisy}}$ based on confidence scores. Formally, let $X_{\mathrm{clean}} = \{x_1, x_2, \ldots, x_{N_c}\}$ be the set of clean flow samples in $\mathcal{D}_{\mathrm{clean}}$, and $y_{\mathrm{clean}} = \{y_1, y_2, \ldots, y_{N_c}\}$ be their corresponding labels. Then, we leverage the supervised learning paradigm to fine-tune our label predictor $f_{\mathrm{prd}}(x)$, which consists of the pre-trained encoder and an added classification head, aiming to minimize the cross-entropy loss: $\mathcal{L}_{\mathrm{CE}} = -\sum_{i=1}^{N_c} y_i \log(f_{\mathrm{prd}}(x_i))$.

Our well-trained label predictor provides confidence scores, $p_i = \max(f\mathrm{prd}(x_i))$, which represent the predictor's certainty regarding the predicted class. These confidence scores serve as the basis for further label correction of noisy flows.

**Noise Label Correction.** To refine the available flows with label noise and avoid losing valuable diversified information, we correct the original labels of noisy flows based on the predicted confidence scores. Specifically, for each noisy flow sample and its label $(x_i, y_i) \in D_{\mathrm{noisy}}$, the predictor outputs a predicted label $\hat{y}_i$ along with its confidence score $p_i = \max(f_{\mathrm{prd}}(x_i))$, which is the highest softmax probability from the label predictor. Based on these confidence scores, we refine the labels of noisy flows into two categories (as examples in Figure 6 of the Appendix B):

(1) **High-Confidence Flow Labels** ($p_i \geq \tau_h$): These samples have high confidence in their predicted labels and are thus assumed to share a similar distribution with the clean set. Thus, we correct their labels with the predicted labels $\hat{y}_i$ and merge them into the refined flows.

(2) **Low-Confidence Flow Labels** ($p_i \leq \tau_l$): These samples have low confidence and likely lie outside the semantic distribution of the clean set. Since they are difficult for the classifier to categorize, we retain their original labels and introduce them to the refined flows to maintain diversity and expand the semantic distribution of samples.

Note that $\tau_h$ and $\tau_l$ represent the thresholds for refining high and low confidence flow labels, respectively. Combined with the clean flows, the obtained refined flows $\mathcal{D}_{\mathrm{refined}}$ contain fewer noisy labels but preserve a more representative semantic distribution of the raw data, including samples that lie outside the initial clean flows's boundaries. The final refined flows are formulated as follows:

$$\mathcal{D}_{\mathrm{refined}} = \mathcal{D}_{\mathrm{clean}} \cup \{(x_i, \hat{y}_i) \mid p_i > \tau_h\} \cup \{(x_i, y_i) \mid p_i < \tau_l\}, \quad (3)$$

where the $\mathcal{D}_{\mathrm{clean}}$ represent clean flows from the semantic-driven noise detector, the $x_i$ is each sample of noisy flows $D_{\mathrm{noisy}}$, $\hat{y}_i$ is the predict label, $y_i$ is the raw label, and $p_i$ is the confidence score.

## 3.3 Cross-Granularity Robust Classifier

Based on the refined flows with low-level label noise, we perform traffic encoder fine-tuning for robust traffic classification. Due to strong fitting capabilities, deep learning models are prone to memorize

mislabeled samples themself, rather than capturing generalizable patterns (43; 44). To avoid the encoder memorizing specific flows with false labels, we propose a cross-granularity robust classifier that integrates both flow-level and packet-level traffic classification tasks. Thus, our classifier can capture generalized patterns that are valid on both tasks, rather than attention to isolated invalid features that are misdirected by noisy labels.

The robust traffic classifier contains two parallel shared weight encoders and classification heads, which could be formulaically treated as a single encoder and denoted as $f_{\text{enc}}$ and $f_{\text{head}}$. The structure and explanation of the shared-weight classifier that can handle both flow and traffic samples are detailed in Appendix C. The encoder is also based on the Transformer and loads the parameters pre-trained by MAE. It first processes each flow $x_i$ by dividing it into sequential packets then selects one packet $x_i^p$ randomly to serve as the other input sample:

$$x_i^p = RandomSelect(\{pac_i^1, pac_i^2, \cdots \mid x_i\}). \tag{4}$$

Next, the flow sample $x_i$ is inputted to the encoder and linear heads to generate $\hat{y}_i$, a flow-level prediction of the label. Following the above process, the flow classification task is performed with the flow sample $x_i$ and label $y_i$ from the refined flows by minimizing the cross-entropy loss. Then, the packet classification task is trained subsequentially during each epoch. Similarly, the randomly selected packet sample is inputted to the encoder and classification head, and obtains the packet level prediction $\hat{y}_i^p$. By associating packet sample $x_i^p$ with the flow label $y_i$ via the cross-entropy loss, the classifier performs packet classification and learns traffic patterns on another granularity.

By the cross-granularity classification tasks, the classifier can utilize both global (flow-level) and local (packet-level) features within the refined flows, providing more varied input against label noise. Specifically, by introducing randomness in the packet selection, the classifier obtains different sample variants in each epoch, thus avoiding the memorization of specific flows with false labels. In addition, the trained cross-granularity robust classifier can serve for both flow-level or packet-level traffic classification under the label noise according to the practical needs, bringing not only robustness but also flexibility.

## 4 Experiments

### 4.1 Experiment Settings

**Datasets.**  We conduct our experiments on four real-world traffic datasets: ISCXVPN (45), Cross-Platform (46), USTC-TFC (16), and Malware (47), each representing different traffic scenarios. Each dataset sample in our experiments corresponds to a network flow, which is obtained via session-aware splitting according to the standard 5-tuple (source IP, destination IP, source port, destination port, and protocol). In FLOWREFINER, we process each flow into a formatted matrix via the MFR algorithm (35). To comprehensively evaluate the robustness of our methods against label noise, we generate noisy datasets for each scenario with different noise ratios $(5\%, 10\%, 20\%, 40\%, 60\%)$. The details of these datasets are shown in Appendix D.

**Baselines.**  To evaluate the performance of FLOWREFINER, we compare it with six state-of-the-art traffic classification methods. These include two traditional ML- and DL-based methods, App-scanner (7) and FS-Net (14); three advanced pre-training methods, ET-BERT (32), MAE (34), and YaTC (35); and the malicious traffic label noise method, MCRe (24). In addition, we introduce three packet classification baselines (48; 16; 32) and six general label noise learning baselines , CE, LSR (49), Mixup (41), GCE (27), SCE (28), Co-teaching (26), and Dividemix (25), to further extend the evaluations.

**Implementation Details.**  In the training stage, we set the batch size as 64, the epochs as 20, and the learning rate as $4*10^{-3}$ with the AdamW optimizer (50). In noise detection, we set the parameter of clustering granularity as $n = 5$, and the top $m = 2$ most frequent labels in the cluster are defined as the majority labels. The high and low confidence thresh- olds are setted as $\tau_h = 0.9$ and $\tau_l = 0.7$. All experiments are implemented in four NVIDIA GeForce RTX3090 GPUs with PyTorch 1.9.0. We summarize the hyperparameter settings in Appendix F.

Table 1: Performance Comparison with Traffic Classification Baselines under Different Noise Ratios.

| Dataset | Noise | Appscanner | | FS-Net | | ET-BERT | | MAE | | YaTC | | MCRe | | Ours | |
|---------|-------|------|------|------|------|------|------|------|------|------|------|------|------|------|------|
| | | Acc | F1 | Acc | F1 | Acc | F1 | Acc | F1 | Acc | F1 | Acc | F1 | Acc | F1 |
| ISCXVPN | 5% | 81.49 | 82.12 | 87.97 | 88.05 | 86.12 | 86.11 | 91.04 | 91.08 | 92.44 | 92.35 | 83.01 | 83.63 | **93.67** | **93.34** |
| | 10% | 79.22 | 79.63 | 85.47 | 85.49 | 85.24 | 85.18 | 89.81 | 89.80 | 89.28 | 89.03 | 81.05 | 81.63 | **91.04** | **90.48** |
| | 20% | 77.27 | 77.95 | 82.57 | 82.57 | 78.91 | 79.07 | 79.26 | 79.29 | 82.60 | 82.49 | 79.88 | 80.64 | **90.86** | **90.15** |
| | 40% | 73.70 | 73.85 | 74.37 | 75.08 | 59.92 | 61.78 | 63.62 | 64.21 | 66.43 | 67.15 | 76.56 | 76.62 | **85.06** | **84.19** |
| | 60% | 57.14 | 57.89 | 56.40 | 58.24 | 34.79 | 36.21 | 42.00 | 43.19 | 43.23 | 45.37 | 69.92 | 70.55 | **73.29** | **72.90** |
| CrossPlatform | 5% | 68.00 | 67.34 | 61.88 | 61.04 | 98.97 | 98.99 | 97.67 | 97.69 | 98.91 | 98.91 | 67.66 | 67.59 | **99.71** | **99.71** |
| | 10% | 66.40 | 65.92 | 59.61 | 59.62 | 98.39 | 98.41 | 95.19 | 95.24 | 98.32 | 98.32 | 63.67 | 63.45 | **99.56** | **99.55** |
| | 20% | 62.53 | 62.39 | 57.89 | 58.07 | 94.46 | 94.49 | 90.67 | 90.75 | 93.51 | 93.58 | 61.48 | 61.14 | **98.76** | **98.75** |
| | 40% | 55.32 | 55.51 | 50.54 | 50.49 | 78.42 | 79.30 | 73.98 | 75.12 | 80.47 | 80.92 | 53.91 | 52.44 | **95.34** | **95.20** |
| | 60% | 43.29 | 44.76 | 39.76 | 41.21 | 56.26 | 59.04 | 48.54 | 50.62 | 57.22 | 58.94 | 43.44 | 40.76 | **84.33** | **84.29** |
| USTC-TFC | 5% | 62.40 | 57.64 | 87.50 | 87.61 | 95.03 | 94.98 | 94.82 | 94.82 | 94.62 | 94.58 | 93.75 | 94.91 | **96.07** | **96.02** |
| | 10% | 63.94 | 60.83 | 86.95 | 86.89 | 93.79 | 93.77 | 94.00 | 93.95 | 93.79 | 93.99 | 93.36 | 94.71 | **95.65** | **95.60** |
| | 20% | 62.65 | 59.95 | 84.76 | 84.70 | 92.33 | 92.34 | 86.13 | 86.46 | 89.86 | 89.75 | 91.02 | 92.91 | **94.20** | **93.93** |
| | 40% | 56.26 | 50.56 | 77.89 | 77.52 | 81.98 | 81.69 | 74.12 | 73.82 | 80.54 | 80.11 | 89.06 | 91.41 | **91.72** | **91.69** |
| | 60% | 51.73 | 50.98 | 70.62 | 70.22 | 58.59 | 58.09 | 47.62 | 48.11 | 56.31 | 56.57 | 75.78 | 75.02 | **78.88** | **78.71** |
| Malware | 5% | 78.16 | 77.86 | 77.81 | 77.90 | 85.95 | 86.01 | 92.57 | 92.59 | 93.11 | 93.11 | 65.62 | 66.77 | **93.38** | **93.33** |
| | 10% | 76.63 | 76.36 | 76.71 | 76.67 | 84.46 | 84.34 | 90.54 | 90.56 | 90.95 | 90.94 | 63.28 | 64.85 | **91.35** | **91.28** |
| | 20% | 76.43 | 76.14 | 75.78 | 75.83 | 77.83 | 78.03 | 81.49 | 81.43 | 83.51 | 83.50 | 62.50 | 64.19 | **88.24** | **88.03** |
| | 40% | 71.26 | 70.67 | 71.40 | 71.08 | 61.35 | 61.49 | 65.27 | 65.39 | 67.97 | 68.49 | 57.23 | 58.83 | **83.51** | **83.33** |
| | 60% | 63.02 | 62.71 | 57.81 | 57.13 | 42.43 | 43.83 | 46.08 | 47.16 | 46.35 | 46.45 | 55.47 | 56.21 | **73.11** | **72.61** |

## 4.2 Comparison with Baselines

**Traffic Classification Baselines.** The performance of our method and other traffic analysis methods on the four traffic datasets under different noise ratios is shown in Table 1. Overall, the performance of all baselines degrades significantly as the label noise rate increases, which highlights the substantial impact of label noise on DL-based methods. Traditional models such as Appscanner and FS-Net are particularly vulnerable, showing rapid performance drops even under moderate noise. Although advanced pre-training methods such as ET-BERT, MAE, and YaTC perform well under low-noise settings such as 5% and 10%, their robustness deteriorates noticeably as the noise level increases. MCRe, which is designed for handling label noise in malicious traffic, shows strong robustness on the USTC-TFC that includes multiple benign and malicious traffic categories. However, its performance degrades significantly on other general classification tasks, such as VPN or mobile app traffic, as well as on the Malware dataset, which contains only recent malware families, due to its reliance on distinct benign-malicious separability.

We can observe that FLOWREFINER consistently outperforms all baselines across datasets and noise levels. Our traffic semantics-driven noise detector and confidence-guided label correction can validly isolate noisy labels and refine the raw flows. Then, the cross-granularity robust classifier can capture meaningful hierarchical semantics and avoid remembering specific noisy flows. As a result, our method consistently achieves accuracy and F1 scores exceeding 70% even at a 60% label noise ratio, where the majority of training samples are incorrectly labeled. Besides, FLOWREFINER also shows better packet-level classification performance compared to the advanced packet classifiers in Appendix G. It can be concluded that our method provides a robust traffic classification framework against label noise to achieve superior performance on various traffic datasets under noisy label conditions.

**Label Noise Learning Baselines.** We further reproduce six general label noise learning methods from other fields on traffic data with consistent encoders for fair comparison, as illustrated in Table 2. LSR and SCE show almost no improvement over CE (i.e., the baseline), while Mixup and GCE can handle label noise relatively effectively due to the interpolating and adaptive loss design. More advanced methods, such as Co-teaching and Dividemix, achieve competitive performance. However, Co-teaching relies on a fixed forgetting rate to select small-loss samples as noise, which could not satisfy various noise conditions. DivideMix, on the other hand, uses a Gaussian mixture model based on loss value distribution to distinguish noise, but it fails when clean or noisy samples are too few to support reliable modeling. In contrast, our method detects noisy flows by modeling traffic semantics rather than relying on loss value distribution, making it more aligned with the nature of traffic data. The results show that FLOWREFINER achieves high performance under different noise ratios and leads the noisy label learning baseline by a large margin.

Table 2: Comparison of F1 Scores with Label Noise Learning Baselines under Different Noise Ratios.

| Method | ISCXVPN | | | | | CrossPlatform | | | | | USTC-TFC | | | | | Malware | | | | |
|---|---|---|---|---|---|---|---|---|---|---|---|---|---|---|---|---|---|---|---|---|
| | 5% | 10% | 20% | 40% | 60% | 5% | 10% | 20% | 40% | 60% | 5% | 10% | 20% | 40% | 60% | 5% | 10% | 20% | 40% | 60% |
| CE | 78.38 | 77.66 | 74.65 | 56.42 | 39.95 | 93.29 | 92.20 | 85.75 | 70.86 | 48.07 | 86.67 | 83.58 | 80.56 | 66.25 | 50.76 | 78.60 | 78.11 | 78.78 | 64.93 | 52.94 |
| LSR | 78.22 | 76.29 | 75.65 | 58.03 | 41.43 | 94.56 | 90.45 | 87.45 | 70.81 | 48.41 | 88.73 | 84.93 | 78.36 | 67.52 | 55.27 | 87.91 | 83.74 | 81.93 | 65.58 | 55.03 |
| Mixup | 79.33 | 77.79 | 79.10 | 66.46 | 48.31 | 90.83 | 89.05 | 86.58 | 78.48 | 61.59 | 91.89 | 90.29 | 87.40 | 73.77 | 64.12 | 89.77 | 88.12 | 82.22 | 73.33 | 62.63 |
| GCE | 75.39 | 74.59 | 71.80 | 65.08 | 49.05 | 93.29 | 92.81 | 92.19 | 90.19 | 79.69 | 82.28 | 80.38 | 78.42 | 77.00 | 68.01 | 86.22 | 81.76 | 80.15 | 74.59 | 64.46 |
| SCE | 77.04 | 73.80 | 73.44 | 59.49 | 40.68 | 94.04 | 93.41 | 87.12 | 76.82 | 60.33 | 86.74 | 85.10 | 85.04 | 73.84 | 56.54 | 88.87 | 83.90 | 77.93 | 67.79 | 56.61 |
| Co-teaching | 78.24 | 84.14 | 87.65 | 76.69 | 55.89 | 82.29 | 87.83 | 92.04 | 83.93 | 62.61 | 86.24 | 88.16 | 81.21 | 81.21 | 57.22 | 86.85 | 88.20 | 84.20 | 80.80 | 60.29 |
| Dividemix | 81.93 | 83.79 | 81.24 | 76.61 | 68.88 | 84.48 | 86.72 | 86.63 | 81.62 | 75.34 | 89.16 | 87.77 | 91.11 | 84.84 | 78.43 | 85.91 | 86.15 | 83.47 | 82.41 | 70.34 |
| Ours | **93.34** | **90.48** | **90.15** | **84.19** | **72.90** | **99.71** | **99.55** | **98.75** | **95.20** | **84.29** | **96.02** | **95.60** | **93.93** | **91.69** | **78.71** | **93.33** | **91.28** | **88.03** | **83.33** | **72.61** |

Table 3: Comparison of F1 Scores on Class-dependent Noise Scenarios.

| Method | ISCXVPN | USTC-TFC |
|---|---|---|
| YaTC | 77.01% | 86.01% |
| MCRe | 76.76% | 80.47% |
| Co-teaching | 76.80% | 76.17% |
| DivideMix | 74.14% | 85.94% |
| Mixup | 69.86% | 78.56% |
| Ours | **80.97%** | **88.40%** |

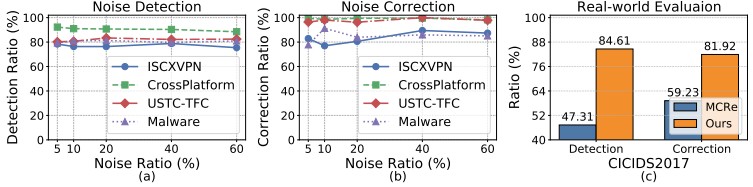

Figure 3: Performance of (a) noise detection, (b) noise correction, and (c) real-world evaluation.

**Class-dependent Noise Evaluation.** Class-dependent noise occurs when the probability of label corruption is not uniform across classes, meaning certain classes are more prone to having mislabeled samples than others. We perform the evaluation on ISCXVPN and USTC-TFC with different ratios of class-dependent noise, where categories with similar patterns are more likely to be mislabeled as each other. The average F1 scores among noise ratios of FLOWREFINER and optimal baselines are shown in Figure 3. The complete results of all baselines and noise ratios are detailed in Appendix H. It can be observed that all methods perform worse compared to the complete random noise setting, since the class-dependent noise leads to more severe category confusion and shifted decision boundaries. Note that our method can still achieve significant performance advantages over the optimal baseline, demonstrating better robustness under different noise types.

## 4.3 Label Noise Evaluation

**Noise Detection Performance.** We evaluate the performance of the traffic semantics-driven noise detector in Figure 3 (a). The detector consistently identifies a large portion of noisy-labeled flows across all datasets, with the detection ratio, i.e., the proportion of detected noise among all noisy samples, remaining above 75% even at 60% noise. This robustness is particularly valuable in security-sensitive tasks such as malware detection, where undetected noise may lead to critical failures. These results confirm the detector's effectiveness in isolating noisy flows and supporting reliable traffic annotation and management.

**Noise Correction Performance.** The confidence-guided label correction is responsible for correcting the labels of the previously selected noisy flows. The correction ratio, i.e. accuracy of the noisy flow correction, is shown in Figure 3 (b). It can be seen that our method can accurately select and assign the correct labels to noisy flows based on prediction confidence, consistently achieving a correction accuracy of over 80% in most scenarios. Particularly on the CrossPlatform and USTC-TFC datasets, labels are almost all correctly assigned to the selected noisy flows, demonstrating the notable performance of the label predictor in this module.

**Real-World Noisy Dataset Evaluation.** Previous studies (20; 21) have revealed significant label noise in the CICIDS2017 dataset (51), and provide a corrected version with revised flow labels. Based on the dataset, we conduct a real-world evaluation, where the training set retains the original noisy labels, and the test set is relabeled according to (20). MCRe, the other traffic label noise method and the only baseline capable of both detecting and correcting label noise, is introduced to the comparison. As shown in Figure 3 (c), our method successfully identifies 84.61% of the noisy flows in training set, significantly outperforming MCRe. Furthermore, the accuracy of our noise detection is 81.92%, achieving high agreement with the expert relabeling.

Table 4: Ablation Study of F1 Scores on The Four Traffic Datasets. The abbreviations are explained as follows: TSND: Traffic Semantics-Driven Noise Detector, CLC: Confidence-Guided Label Correction Module, CRC: Cross-Granularity Robust Classifier.

| Method | ISCXVPN | | | | | CrossPlatform | | | | | USTC-TFC | | | | | Malware | | | | |
|---|---|---|---|---|---|---|---|---|---|---|---|---|---|---|---|---|---|---|---|---|
| | 5% | 10% | 20% | 40% | 60% | 5% | 10% | 20% | 40% | 60% | 5% | 10% | 20% | 40% | 60% | 5% | 10% | 20% | 40% | 60% |
| Ours | 93.34 | 90.48 | 90.15 | 84.19 | 72.90 | 99.71 | 99.55 | 98.75 | 95.20 | 84.29 | 96.02 | 95.60 | 93.93 | 91.69 | 78.71 | 93.33 | 91.28 | 88.03 | 83.33 | 72.61 |
| w/o TSND | 93.00 | 88.99 | 84.10 | 64.27 | 48.60 | 97.69 | 95.93 | 88.52 | 68.84 | 46.85 | 95.99 | 93.87 | 91.43 | 75.66 | 48.95 | 93.14 | 92.27 | 84.14 | 67.65 | 47.42 |
| w/o CLC | 88.85 | 85.09 | 80.12 | 77.76 | 69.71 | 92.81 | 90.84 | 89.69 | 86.35 | 78.01 | 94.88 | 93.26 | 92.78 | 89.70 | 77.59 | 80.32 | 79.15 | 79.11 | 74.78 | 65.29 |
| w/o CRC | 91.33 | 88.39 | 85.05 | 78.42 | 65.16 | 97.99 | 98.07 | 95.41 | 88.23 | 70.04 | 95.55 | 95.40 | 93.38 | 90.01 | 74.60 | 86.95 | 89.56 | 83.85 | 78.68 | 63.87 |

## 4.4 Ablation Study

The ablation study is conducted to evaluate the contribution of each component in FLOWREFINER. As shown in Table 4, the performance declines consistently on all traffic datasets when any of the key components is removed. In particular, the removal of the traffic semantics-driven noise detector (TSND) results in the most severe performance degradation under high noise ratios. For example, on the ISCXVPN dataset with 60% noise, removing TSND leads to a significant drop in accuracy from 73.29% to 47.62%. Similarly, on the CrossPlatform dataset, accuracy drops from 84.33% to 44.89%. This highlights the critical role of the TSND in accurately identifying and isolating noisy flows, which ensures that the subsequent components operate with more reliable data. When the confidence-guided label correction (CLC) is ablated, the performance also degrades, suggesting that this component can effectively against label noise by handling difficult cases and expanding the clean set. Note that our method has less performance degradation on the USTC-TFC dataset compared to other datasets under the ablation of CLC. This is because our noise detector has detected most of the noisy flows in USTC-TFC according to Figure 3, offering a highly clean flow set for traffic classifier training. Finally, the removal of the structure of cross-granularity robust classifier (CRC) results in stable performance reductions under all datasets and noise ratios. It proves that the joint task of both flow and packet classification could improve the robustness under different noise ratios.

## 4.5 Discussions

**The Impact of Parameters.** We investigate the effect of two key parameters in our noise detection module: the clustering granularity $n$ and the majority label count. The clustering granularity $n$ defines the total number of clusters $K = n \times C$, where $C$ is the number of classes. It controls how finely label-independent semantics are modeled. As shown in Figure 4, on the ISCXVPN dataset, increasing $n$ from 1 to 5 notably improves the F1 score. A setting of $n = 5$ achieves optimal performance by balancing semantic sensitivity and robustness, whereas higher values (e.g., $n = 7$) lead to over-segmentation and misclassification of clean samples as noise. The majority label count determines how many of the most frequent labels in each cluster are retained as clean. Using only the top-1 label (count = 1) can misclassify semantically similar clean flows as noise, while larger counts may retain actual noisy samples. As shown in Figure 4, a count of 2 provides the best trade-off across noise ratios by tolerating intra-cluster semantic variation without sacrificing detection precision. Overall, both parameters play a critical role in balancing noise detection sensitivity and robustness, with $n = 5$ and majority label count = 2 yielding consistently strong performance across settings.

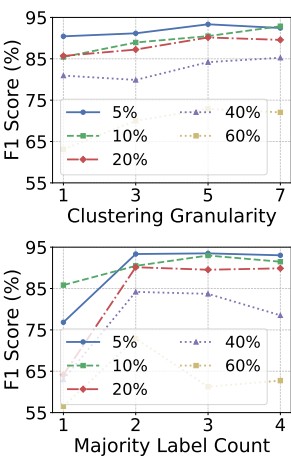

Figure 4: Parameters Impact.

**Limitations.** As a deep learning-based method, FLOWREFINER requires substantial computational resources for efficient training, preferably with GPU acceleration. Our framework currently takes about 3 minutes on an RTX 3090 GPU with 2.79 GB of memory for training. In future work, we will continue to optimize the training pipeline to reduce computational overhead and improve accessibility, while also leveraging ongoing progress in lightweight inference and dedicated hardware accelerators, which are expected to further enhance the practicality and scalability of FLOWREFINER in real-world deployments.

## 5  Conclusion

In this paper, we proposed FLOWREFINER, a robust framework for general traffic classification under noisy label conditions. Our method addresses the significant challenges of label noise, including its detection, correction, and robust classification, thus reducing dependency on high-quality labeled data. Through the integration of a semantic-driven noise detector, confidence-guided label correction, and a cross-granularity robust classifier, FLOWREFINER effectively leverages noisy traffic to improve classification performance. Our experimental results across various datasets and scenarios demonstrate the ability of FLOWREFINER to outperform state-of-the-art methods in both accuracy and resilience to noise. This framework provides a valuable solution for enhancing general traffic analysis, particularly in environments where high-quality labels are difficult to obtain, and sets the stage for future advancements in traffic classification under noisy conditions.

## 6  Acknowledgement

The authors thank the anonymous reviewers for their valuable comments and suggestions that helped improve this paper. This work was supported in part by China National Funds for Distinguished Young Scientists under Grant 62425201; in part by the National Key Research and Development Program of China under Grant 2023YFB3107100; in part by the National Natural Science Foundation of China under Grant 62502089, Grant 62172093, Grant 62132011, and Grant U22B2025; in part by Basic Research Program of Jiangsu under Grant BK20251353; in part by Fundamental and Interdisciplinary Disciplines Breakthrough Plan of the Ministry of Education of China under Grant JYB2025XDXM114; and in part by Beijing-Tianjin-Hebei Natural Science Foundation Cooperation Project under Grant 25JJJJC0003.

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

## A Building of the Traffic Semantics-oriented Encoder

Figure 5 depicts the process of building the traffic semantics-oriented encoder with the pre-training paradigm of a masked autoencoder (MAE). Initially, the network traffic data captured from Pcap files undergoes preprocessing to extract and resize the flow content, ensuring the data is in a suitable format for model training. The processed traffic data is then intentionally masked, and input into the traffic encoder, which employs the principles of an MAE by aiming to recover the original data from its corrupted form. The encoder learns to extract robust feature representations by focusing on the structure left intact by the masking process. Finally, the encoded features are passed through a decoder that attempts to reconstruct the original input, thereby enabling the model to learn critical data characteristics effectively and allowing the encoder to extract effective traffic semantics.

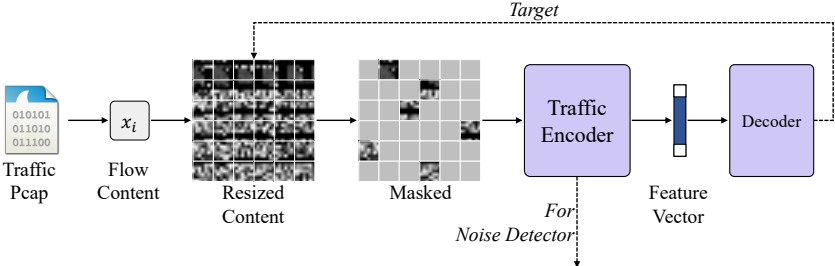

Figure 5: The pre-training paradigm of masked autoencoder for building traffic semantics-oriented encoder.

## B Two Categories of the Predicted Labels from the Traffic Label Predictor

Based on the label predictor fine-tuned on the clean flows, we could obtain the predicted labels of detected noisy flows. As shown in Figure 6, we refine the labels of noisy flows into two categories, i.e., high-confidence flow labels and low-confidence flow labels. The samples with high-confidence labels share a similar distribution with the clean set, and we use the prediction results to correct their labels. The samples with low-confidence flow labels represent difficult instances for the predictor. Thus, we retain their original labels and introduce them to the refined flows to maintain diversity and expand the semantic distribution of samples.

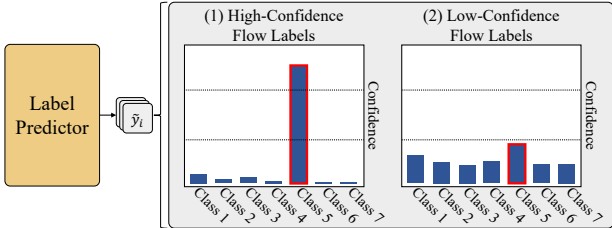

Figure 6: Two categories of the predicted labels from the traffic label predictor.

## C The Structure Detail of the Cross-Granularity Robust Classifier

Figure 7 presents the structure of the Cross-Granularity Robust Classifier, which is designed to classify network traffic at both the packet and flow levels. In the flow-level traffic classifier, multiple parallel packet encoders process each individual packet from the flow. These encoders share weights,

indicating that they operate under a unified framework to maintain consistency in feature extraction across different packets. The encoded packet features are then aggregated to form a comprehensive representation of the flow, which is subsequently used to perform the flow-level traffic classification for the entire flow. This structure of parallel packet encoders of the flow-level classifier enables a share-weight individual packet encoder to process packet-level input and thus perform packet-level classification. In detail, a packet is selected randomly from the flow sample, and then classified by the packet-level traffic classifier. This dual approach allows the system to leverage fine-grained packet-level data along with aggregated flow-level information, enhancing the robustness and accuracy of the traffic classification.

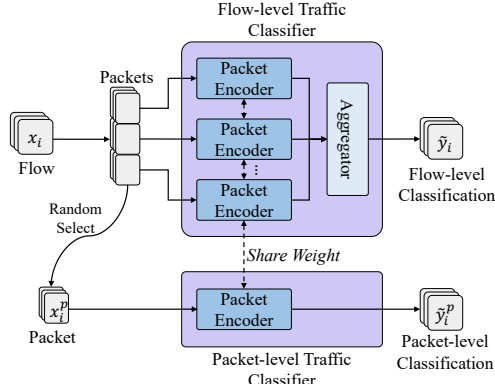

Figure 7: The structure detail of the cross-granularity robust classifier.

## D  Dataset Information

The details of the four real-world traffic datasets used in this work are shown as follows:

- **ISCXVPN** (45) contains 7 categories of encrypted traffic, collected by the Canadian Institute for Cybersecurity via OpenVPN. The dataset consists of 2,275 training samples and 569 test samples.
- **CrossPlatform** (46) includes encrypted traffic data from various platforms. In our experiments, we use IOS application traffic in the China region, comprising 30 different categories. The dataset consists of 5,429 training samples and 1,372 test samples.
- **USTC-TFC** (16) is a traffic dataset with 10 categories of benign application traffic and 10 categories of malware traffic. The dataset consists of 1,914 training samples and 483 test samples.
- **Malware** (47) is a recently published dataset featuring traffic from 10 malware families, available at `https://malware-traffic-analysis.net/about.html`. The dataset consists of 2,938 training samples and 740 test samples.

## E  Baselines

To evaluate the performance of our method, we use six state-of-the-art traffic classification methods and five label noise learning methods as baselines. The following traffic classification baselines include one traditional machine learning method, three traditional deep learning methods, and three advanced pre-training methods.

- **Appscanner** (7): A traditional machine learning tool that classifies traffic using handcrafted flow-based features, often less resilient to noisy or encrypted data.
- **FS-Net** (14): A flow-based model incorporating both classification and reconstruction tasks, enhancing its robustness to noise and improving semantic feature extraction.
- **ET-BERT** (32): A BERT-like Transformer pre-trained model, leveraging self-supervised learning to capture flow-level semantic patterns, offering strong resilience to label noise.
- **MAE** (34): A self-supervised traffic classifier based on Masked Autoencoder (MAE), masking parts of traffic flows and reconstructing them, which makes it robust in noisy conditions.
- **YaTC** (35): A hybrid traffic classification model combining supervised and self-supervised learning, balancing noise handling and effective traffic feature extraction.
- **MCRe** (24): A state-of-the-art traffic analysis method focuses on malicious traffic label noise learning.

In addition, we introduce the following label noise learning baselines, all of which use the same encoder structure for fair comparison.

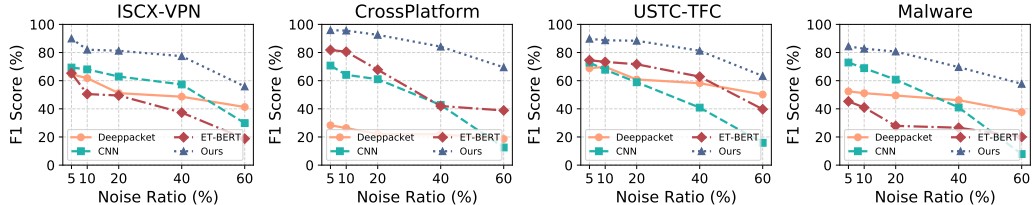

Figure 8: Comparison of packet classification performance with packet-level baselines.

- **CE**: A standard cross-entropy loss used in classification tasks, which can be sensitive to label noise as it directly penalizes incorrect predictions.
- **LSR** (49): Label Smoothing Regularization reduces model overconfidence by distributing probability mass to incorrect labels, making the model more robust to noise.
- **Mixup** (41): A data augmentation technique that generates synthetic samples by interpolating between two samples and their labels, helping the model generalize better under noisy conditions.
- **GCE** (27): Generalized Cross-Entropy blends the advantages of MAE and CE to address label noise, making it more robust by adjusting the loss function dynamically.
- **SCE** (28): Symmetric Cross-Entropy introduces a correction term to balance the standard CE loss, aiming to better handle noisy labels by mitigating over-penalization of incorrect predictions.
- **Co-teaching** (26): Trains two networks simultaneously and lets them teach each other by selecting small-loss samples, assuming that clean samples have lower loss values. This helps filter out noisy labels during training.
- **DivideMix** (25): Uses a Gaussian mixture model to divide samples into clean and noisy sets based on loss values, then applies semi-supervised learning to train the model using both labeled and unlabeled data.

## F   Hyperparameter Information

In all experiments, we use a batch size of 64, train for 20 epochs, and optimize with AdamW (50) using a learning rate of $4 \times 10^{-3}$. For noise detection, we set the clustering granularity to $n = 5$, with the top $m = 2$ most frequent labels in each cluster considered as majority labels. Confidence thresholds are set at $\tau_h = 0.9$ and $\tau_l = 0.7$ to balance precision and recall in noise filtering. Importantly, we apply the same hyperparameter settings across all datasets, demonstrating that our method achieves strong performance without extensive tuning, highlighting its robustness to hyperparameter choices.

Table 5: Hyperparameter settings in our experiments.

| Parameter | Setting |
| --- | --- |
| Batch size | 64 |
| Epochs | 20 |
| Learning rate | $4 \times 10^{-3}$ |
| Optimizer | AdamW (50) |
| Clustering granularity ($n$) | 5 |
| Majority label count ($m$) | 2 |
| High confidence threshold ($\tau_h$) | 0.9 |
| Low confidence threshold ($\tau_l$) | 0.7 |

## G   Packet Classification Ability

The proposed cross-granularity robust classifier integrates both flow-level and packet-level traffic classification tasks against label noise, bringing the ability to serve as a robust packet classifier. We compare the packet-level classification performance with advanced packet classifiers, including Deeppacket (48), CNN (16), and ET-BERT (32). As shown in Figure 8, in packet-level traffic analysis

Table 6: Comparison with Traffic Classification Baselines on Class-dependent Noise Scenarios.

| Dataset | Noise | Appscanner | | FS-Net | | ET-BERT | | MAE | | YaTC | | MCRe | | Ours | |
|---|---|---|---|---|---|---|---|---|---|---|---|---|---|---|---|
| | | Acc | F1 | Acc | F1 | Acc | F1 | Acc | F1 | Acc | F1 | Acc | F1 | Acc | F1 |
| ISCXVPN | 5% | 81.49 | 82.07 | 87.42 | 87.58 | 85.76 | 85.77 | 91.39 | 91.36 | 93.15 | 92.98 | 80.86 | 81.72 | **93.32** | **92.98** |
| | 10% | 81.16 | 81.70 | 85.78 | 86.21 | 81.90 | 81.69 | 90.51 | 90.45 | 91.21 | 91.13 | 78.52 | 79.29 | **92.62** | **92.28** |
| | 20% | 80.51 | 80.36 | 83.04 | 83.33 | 80.49 | 80.52 | 84.35 | 84.44 | 88.75 | 88.64 | 77.54 | 77.38 | **89.45** | **88.90** |
| | 40% | 67.20 | 68.15 | 69.14 | 70.29 | 55.18 | 55.99 | 68.71 | 69.50 | 67.31 | 68.28 | 74.61 | 74.75 | **78.03** | **77.29** |
| | 60% | 49.67 | 51.14 | 47.10 | 47.95 | 46.04 | 48.23 | 46.74 | 48.35 | 42.00 | 44.03 | 45.51 | 46.29 | **52.72** | **53.38** |
| USTC-TFC | 5% | 62.14 | 62.15 | 86.33 | 86.53 | 94.20 | 94.18 | 93.58 | 93.52 | 95.24 | 94.20 | 94.12 | 94.25 | **94.62** | **94.50** |
| | 10% | 61.38 | 61.38 | 85.78 | 85.98 | 92.96 | 93.07 | 93.79 | 93.75 | 93.17 | 93.18 | 90.23 | 92.40 | **94.20** | **94.16** |
| | 20% | 60.86 | 55.61 | 80.39 | 79.97 | 91.51 | 91.52 | 88.61 | 88.62 | 90.68 | 90.65 | 89.45 | 91.80 | **92.13** | **92.15** |
| | 40% | 54.98 | 50.17 | 76.56 | 77.01 | 75.98 | 76.41 | 78.88 | 79.99 | 83.43 | 83.73 | 82.03 | 83.87 | **84.47** | **84.90** |
| | 60% | 42.19 | 41.75 | 62.34 | 62.52 | 54.65 | 57.11 | 56.10 | 57.27 | 66.87 | 68.28 | 35.55 | 40.02 | **75.36** | **76.30** |

Table 7: Comparison with Label Noise Learning Baselines on Class-dependent Noise Scenarios.

| Dataset | Noise | CE | | LSR | | Mixup | | GCE | | SCE | | Co-teaching | | DividMix | | Ours | |
|---|---|---|---|---|---|---|---|---|---|---|---|---|---|---|---|---|---|
| | | Acc | F1 | Acc | F1 | Acc | F1 | Acc | F1 | Acc | F1 | Acc | F1 | Acc | F1 | Acc | F1 |
| ISCXVPN | 5% | 78.91 | 79.06 | 79.61 | 79.66 | 82.95 | 82.74 | 76.09 | 74.58 | 79.09 | 79.06 | 85.41 | 84.44 | 85.24 | 85.37 | **93.32** | **92.98** |
| | 10% | 76.09 | 75.31 | 77.85 | 77.34 | 78.91 | 78.76 | 75.04 | 73.88 | 77.50 | 76.47 | 86.46 | 85.74 | 80.84 | 81.37 | **92.62** | **92.28** |
| | 20% | 76.44 | 76.12 | 74.34 | 74.02 | 76.80 | 76.67 | 72.93 | 71.89 | 74.34 | 74.77 | 87.69 | 87.37 | 79.61 | 80.15 | **89.45** | **88.90** |
| | 40% | 58.52 | 59.16 | 59.75 | 59.91 | 67.48 | 67.39 | 68.18 | 67.23 | 64.49 | 64.37 | 75.57 | 75.50 | 76.63 | 77.00 | **78.03** | **77.29** |
| | 60% | 38.84 | 40.05 | 34.62 | 36.01 | 43.76 | 45.73 | 42.88 | 42.03 | 41.82 | 44.02 | 51.85 | 50.94 | 50.44 | 53.44 | **53.25** | **53.62** |
| USTC-TFC | 5% | 87.16 | 86.73 | 86.33 | 85.67 | 90.68 | 90.45 | 80.12 | 78.79 | 86.75 | 86.45 | 86.81 | 73.84 | 85.92 | 85.72 | **94.62** | **94.50** |
| | 10% | 85.92 | 86.06 | 83.85 | 83.31 | 89.23 | 88.51 | 79.50 | 78.66 | 85.92 | 85.86 | 77.64 | 74.63 | 90.48 | 90.41 | **94.20** | **94.16** |
| | 20% | 81.78 | 81.27 | 82.81 | 82.27 | 83.22 | 82.49 | 78.67 | 76.84 | 80.12 | 79.90 | 83.44 | 80.90 | 89.23 | 88.98 | **92.13** | **92.15** |
| | 40% | 65.21 | 65.18 | 66.04 | 66.20 | 74.12 | 74.41 | 71.42 | 70.16 | 71.22 | 70.62 | 82.40 | 81.75 | 81.78 | 82.25 | **84.47** | **84.90** |
| | 60% | 58.17 | 58.86 | 51.55 | 51.65 | 55.69 | 56.93 | 64.80 | 65.05 | 48.44 | 49.47 | 68.12 | 69.73 | 65.22 | 66.35 | **75.36** | **76.30** |

tasks, where each sample contains less information, the classification methods suffer from label noise more. Only FLOWREFINER can achieve F1 scores higher than 80% under 20% noise ratio, while other packet-level baselines can hardly achieve 60%. It demonstrates that FLOWREFINER could obtain two levels of robust encrypted traffic classifier at once and adapt to different requirements.

# H  Class-dependent Noise Evaluation

Class-dependent noise occurs when the probability of label corruption is not uniform across classes, meaning certain classes are more prone to having mislabeled samples than others.

For instance, in VPN traffic categories, VOIP and Streaming are often mislabeled due to their similar real-time transmission metrics, such as packet sizes and intervals, once encrypted. FTP and MAIL, which exhibit prolonged, high-volume traffic traits during large file transfers, are frequently miscategorized as p2p. Likewise, BROWSING and CHAT, with their frequent interactions and small packet sizes, become indistinguishable when encrypted, leading to frequent mislabeling. Thus we injected class-dependent noise into the ISCXVPN dataset to mirror these common mislabeling scenarios.

On the other hand, in the USTC-TFC malware traffic dataset, FTP and SMB often get confused due to their similar file transfer behaviors. Both protocols involve significant data packet exchanges which can appear alike under traffic analysis. Similarly, Gmail and Outlook, both being email services, often exhibit interchangeable traffic patterns due to similar data flow structures, leading to potential mislabeling. Services like Skype and Facetime, which both facilitate VoIP communications, show closely related network signatures that can easily be mistaken for one another when encrypted. Additionally, applications like WorldOfWarcraft and Skype may be misclassified due to their real-time interaction requirements, which create similar traffic spikes. BitTorrent, known for peer-to-peer file sharing, and FTP, used for direct file transfers, also share large file movement characteristics that can be confused under automated analysis. Furthermore, malware such as Zeus and Cridex, or Nsis-ay and Virut, exhibit overlapping behaviors in terms of their network communication patterns, making accurate classification challenging. Thus, we injected these class-dependent noises into the USTC-TFC dataset to closely simulate these realistic mislabeling scenarios.

We comprehensively performed a comparison with both traffic classification methods and label noise methods under class-dependent noise, which are shown in Table 6 and Table 7. Results show that our method can achieve significant performance advantages over the baselines in the class-dependent noise scenario.

