# OpenReview forum: "FlowRefiner: A Robust Traffic Classification Framework against Label Noise"
_NeurIPS.cc/2025/Conference — NeurIPS 2025 poster_

### Official Review · Reviewer_fb8o · 2025-06-07

**Clarity:** 4
**Significance:** 2
**Originality:** 2
**Rating:** 4
**Confidence:** 5

**Summary:**

This paper proposes FlowRefiner, a three-stage framework for network traffic classification in the presence of label noise. The framework first uses a "traffic semantics-driven noise detector," which leverages a pre-trained Transformer-based encoder to generate embeddings and then applies clustering to identify noisy samples. Second, a "confidence-guided label correction" module refines the labels of suspected noisy flows. Finally, a "cross-granularity robust classifier" is trained on the cleaned data, using both flow-level and packet-level information to improve robustness. The authors conduct an extensive experimental evaluation on four datasets, demonstrating that their method significantly outperforms several existing baselines under various types of synthetic and real-world label noise.

**Questions:**

1. Did you use a session-aware or time-based splitting strategy to avoid leakage? If not, could you re-evaluate your results with a more robust approach?
2. Can you provide stronger theoretical or empirical justification for the choice that keeps the low-confidence samples? Have you compared this against simply removing or correcting such samples?
3. If possible, could you please add related methods like RAPIER and ULDC as baselines?
4. Could you more clearly distinguish your contributions from existing literature? What specific insights or components in FlowRefiner are novel beyond architectural substitutions?

**Ethical Concerns:**

["NO or VERY MINOR ethics concerns only"]

**Final Justification:**

My score has increased because the authors have proven the technical soundness and experimental correctness of their work. However, my recommendation is not a strong acceptance because the central question of the paper's positioning and its alignment with a fundamental methodology conference like NeurIPS remains. The work is a high-quality contribution, but its impact may be better appreciated at a top-tier applied venue.

**Limitations:**

yes

**Paper Formatting Concerns:**

No formatting issue detected.

**Quality:**

3

**Strengths And Weaknesses:**

The paper is well-structured, clearly written, and addresses a practical and important problem in the domain of network security and management: the prevalence and impact of label noise on traffic classification.

### **Strengths**
1. **Clarity and Quality of Presentation:** The paper is exceptionally well-written. The proposed FlowRefiner framework is described in a clear, step-by-step manner, and the figures are informative. The overall presentation quality is high.
2. **Thorough Experimental Evaluation:** The authors have conducted a comprehensive set of experiments across multiple datasets, various noise types (symmetric, asymmetric, class-dependent), and high noise ratios. The ablation study effectively demonstrates the contribution of each component of the framework. The performance gains shown over the chosen baselines are indeed substantial.
3. **Problem Significance (for its domain):** Label noise is a genuine challenge for applying deep learning to real-world network traffic data. A robust framework to mitigate this issue is a valuable contribution to the applied networking and security community.

### **Weaknesses**
1. **Lack of Originality:** The primary weakness of this paper is the overstatement of its novelty. The core technical ideas are direct applications or minor variations of well-established techniques in the label noise and traffic analysis literature:
    - The "traffic semantics-driven noise detector" is conceptually identical to prior work that uses unsupervised pre-training to learn embeddings and then performs outlier detection in the feature space via clustering or density estimation. For instance, frameworks like RAPIER [1] and ULDC [2] employ a nearly identical pipeline of autoencoder-based feature extraction followed by distribution-based confidence scoring to identify noisy labels. The use of a Transformer encoder (MAE) is a modern architectural choice, but it does not change the fundamental, pre-existing concept.
    - The idea of using cross-granularity (flow and packet level) information for robustness is also not new. Prior work has explored multi-granularity and hierarchical approaches in traffic classification with the explicit motivation of improving robustness to noise and variations [3]. While the specific implementation might differ, the core idea is not a novel contribution.
2. **Lack of Experimental Comparison**: The paper seems to omit the most relevant baselines. The authors should have compared FlowRefiner against other end-to-end noisy label frameworks for traffic, such as the aforementioned RAPIER and ULDC.
3. **Questionable Methodological Choices:**
    - **Handling of Low-Confidence Samples.** The framework identifies a set of likely noisy flows, and for samples within this set where the label predictor has low confidence, the authors opt to "retain their original labels" in order to "maintain diversity." While the intention to preserve diversity is understandable, this decision raises important concerns. Retaining samples that are likely mislabeled and reintroducing them into the training set could inadvertently compromise the effectiveness of the data cleaning process and potentially degrade the performance of the final classifier. The rationale for maintaining diversity would benefit from stronger theoretical grounding or empirical evidence to justify this trade-off.
    - **Insufficient Detail and Potential Data Leakage.** The description of the data preparation process would benefit from additional detail, as its current form raises questions about the validity of the reported results. While the paper provides the sizes of the training and testing sets, it does not clarify how these splits were performed. This is particularly important for network traffic data, which often exhibits strong temporal and session-based dependencies. A simple random split of flows or packets could inadvertently lead to data leakage, for example, when data from the same session appears in both training and test sets, potentially leading to overestimated performance metrics. It would strengthen the paper significantly if the authors clarified whether a more robust splitting strategy (such as by time or by user/host) was employed.

Overall, while the paper tackles an important problem in network security and presents a technically sound solution, its contributions are largely incremental and grounded in well-established methods. The work offers more of an engineering refinement than a fundamental advancement in machine learning. As such, it may have limited appeal to the broader NeurIPS community and would be better suited for a more specialized venue.

[1] Qing, Y., Yin, Q., Deng, X., Chen, Y., Liu, Z., Sun, K., ... & Li, Q. (2023). Low-quality training data only? A robust framework for detecting encrypted malicious network traffic. arXiv preprint arXiv:2309.04798.

[2] Yuan, Q., Zhu, Y., Xiong, G., Wang, Y., Yu, W., Lu, B., & Gou, G. (2024). ULDC: Unsupervised learning-based data cleaning for malicious traffic with high noise. The Computer Journal, 67(3), 976-987.

[3] Tang, P., Dong, Y., Mao, S., Wei, H. L., & Jin, J. (2023). Online classification of network traffic based on granular computing. IEEE Transactions on Systems, Man, and Cybernetics: Systems, 53(8), 5199-5211.

---

> ### Author Rebuttal · Authors · 2025-07-31
>
> Thank you for your constructive feedback and thoughtful suggestions. We hope that our detailed responses below will address your concerns.
>
> ### **About Originality.**
>
> > W1: The core technical ideas are direct applications or minor variations of well-established techniques in the label noise and traffic analysis literature
>
>
> We respectfully clarify that FlowRefiner is not a 'direct application or minor variation' of existing methods, but rather the **first work tackling label noise in general and complex traffic scenarios**, a challenge that has not been addressed in previous literatures.
>
> > **Q4**: Could you more clearly distinguish your contributions from existing literature? What specific insights or components in FlowRefiner are novel beyond architectural substitutions?
>
>
> Below, we further clarify our fundamental novelty and distinct contribution compared to prior works such as [1][2][3], in terms of insights, methodology, and differences:
>
>
> ### **About RAPIER [1] and ULDC [2].**
>
> > W1.1: The "traffic semantics-driven noise detector" is conceptually identical to prior work ... like RAPIER [1] and ULDC [2]
>
> The mentioned RAPIER and ULDC, which **are already cited** ([21,37]) and **discussed** in our Introduction (lines 44–51) and Related Work (lines 101–104).
>
> ### **Insights Beyond [1][2].**
>
>
> Specifically, previous methods (e.g., RAPIER, ULDC) **solely rely on samples themself** (density or distance) to determine "real label" with **simplified assumptions**:
>  - RAPIER **requires clear density differences between normal and malicious** traffic, binary relabeling samples in densest and sparsest regions accordingly.
>  - ULDC defines a single center for each class and considers distant samples as noisy, which **assumes samples from different classes are obviously separable**.
>
> However, our core insight is that **general traffic classification tasks are inherently more complex than these simplified assumptions**. As illustrated in our manuscript (lines 156-159 and Figure 2):
>
>  - **Same class** samples often exhibit **distinct** semantics, e.g., MAIL traffic covers both large attachment transfers and quick, small email exchanges with behavior diversity.
>  - **Different class** samples may appear semantically **similar**, e.g., both FTP and p2p traffic transfer files and therefore produce highly similar behaviors.
>
> ### **Methodology Beyond [1][2].**
>
> Motivated by this critical insight, where **real labels cannot be reliably determined by sample features alone**, FlowRefiner introduces the following key novelties:
>
>  - **Sematics extraction**: we introduce pre-trained encoder to extract **high-dimentional semantics**, which can validly represent traffic behaviors. In contrast, RAPIER and ULDC use AE only to reduce the dimension of the feature for density or distance calculations.
>  - **Noise detection**: we innovatively utilize **label distribution** within semantic clusters to detect label noise, going beyond previous works that identify labels merely via sample distance or density.
>  - **Noise correction**: we further refine detected noisy flows according to the  **confidence of the predictor**, which is fine-tuned by clean flows to correct detected noisy samples. In comparison, the 'confidence' of ULDC is simply defined by sample distance.
>
>
> ### **Differences from [3].**
>
> > W1.2: Prior work has explored multi-granularity ... with the explicit motivation of improving robustness to noise and variations [3].
>
> We clarify the **irrelevance** of [3] and FlowRefiner, as the **definitions of both noise and granularity** in [3] are completely **different** from ours**:
>
>
> |          | Noise Definition | Granularities |
> | - | - | - |
> | [3]         | **Network noise**: packet loss, retransmission, and disorder in dynamic networks | Spatial granule and temporal granule: **neighborhood packets** with similar packet size and interarrival time |
> | FlowRefiner | **Label noise**: incorrectly labeled samples                 | Flow-level and packet-level: two granularities of **classification tasks** in traffic analysis |
>
>
> - Noise in [3] refers to packet loss, retransmission, and disorder in dynamic networks, which is irrelevant to our label noise issues.
> - [3] defines spatial granule and temporal granule as neighborhood packets with similar packet size and interarrival time separately. In contrast, our cross-granularity describes the flow- and packet-level classifier.
>
>
>
>
> As detailed above, we firmly believe these explicit distinctions clearly establish FlowRefiner’s fundamental novelty and significant practical contributions, distinctly differentiating it from prior works, e.g., [1],[2],[3].
>
> ### **Lack of Experimental Comparison.**
>
> > W2: The paper seems to omit the most relevant baselines. The authors should have compared FlowRefiner against other end-to-end noisy label frameworks for traffic.
>
> As stated in Section 4.1 and Appendix E, we included **MCRe**, an **end-to-end** malicious traffic label noise method, as the **most relevant baseline**. Additionally, ULDC claims itself as a data cleaning method instead of an end-to-end method.
>
>
> > **Q3**: If possible, could you please add related methods like RAPIER and ULDC as baselines?
>
> - RAPIER **cannot support multi-classification tasks methodologically**, thus it is not applicable to our general evaluation.
> - ULDC did not release their source code, but we compared **MCRe[22], which is the follow-up work of the same authors that proposed ULDC**, in Section 4. And the MCRe paper compared ULDC and achieved better performance.
>
> Nevertheless, we further reproduced ULDC, and the results are as follows:
>
>
>
> | ISCXVPN | 5% | 10%       | 20%       | 40%       | 60%       |
> | - | - | - | - | - | - |
> | ULDC  | 48.57     | 40.53     | 39.84     | 39.27     | 38.36     |
> | ULDC wo noise clean | 73.09     | 72.68     | 70.88     | 69.33     | 62.60     |
> | MCRe | 83.63     | 81.63     | 80.64     | 76.62     | 70.55     |
> | Ours  | **93.34** | **90.48** | **90.15** | **84.19** | **72.90** |
>
> | USTC-TFC | 5%        | 10%       | 20%       | 40%       | 60%       |
> | - | - | - | - | - | - |
> | ULDC | 69.94   | 67.73   | 66.12     | 65.65     | 62.27     |
> | ULDC wo noise clean | 90.30     | 84.94     | 79.06     | 78.04     | 65.95     |
> | MCRe | 94.91     | 94.71     | 92.91     | 91.41     | 75.02     |
> | Ours | **96.02** | **95.60** | **93.93** | **91.69** | **78.71** |
>
>
>
> The results demonstrate that both MCRe and FlowRefiner outperform ULDC. ULDC can not handle label noise because its simple noise cleaning assumption cannot adapt to general scenarios.
>
>
>
> ### **Handling of Low-Confidence Samples.**
>
> > W3.1 and **Q2**: Can you provide stronger theoretical or empirical justification for the choice that keeps the low-confidence samples? Have you compared this against simply removing or correcting such samples?
>
> We further provide the F1 scores under different processes of low-confidence samples:
>
> |ISCXVPN|5%|10%|20%|40%|60%|
> | - | - | - | - | - | - |
> | correcting low-confidence samples| 89.32 | 83.99 | 81.57 | 79.31 | 69.59 |
> | removing low-confidence samples  | 90.56 | 86.00 | 81.98 | 80.39 | 71.00 |
> | Ours            | **93.34** | **90.48** | **90.15** | **84.19** | **72.90** |
>
> The results show that removing or correcting low-confidence samples consistently leads to worse performance. Since such samples are few and ambiguous, keeping their original labels has a limited negative impact but preserves diversity and boundary coverage that benefits model robustness.
>
>
> ### **Insufficient Detail.**
>
> > W3.2 and **Q1**: Did you use a session-aware or time-based splitting strategy to avoid leakage? If not, could you re-evaluate your results with a more robust approach?
>
> We perform strict session-aware splitting to avoid leakage, where each flow corresponds to one sample. Even in the evaluation of packet classification, we still ensure each packet is sampled from a unique flow.
> This strict setting of traffic splitting avoids potential data leakage with unexpected dependencies, and ensures a fair and robust evaluation of our method’s performance.
> We will clarify these details in the revision.

---

> > ### Comment · Reviewer_fb8o · 2025-08-03
> >
> > The rebuttal effectively addresses my concerns on experimental validation with new data. However, my reservation about the work's fundamental methodological novelty for NeurIPS remains, as the contribution is more of a sophisticated application of existing techniques.
> >
> > Given the improved rigor, I will raise my score, but still believe the paper may be better suited for a top-tier applied conference.

---

> > > ### Author Response · Authors · 2025-08-05
> > >
> > > Thank you for your prompt feedback and for increasing the score. We sincerely appreciate your time and effort.
> > >
> > > Our work tackles the challenge of label noise in network traffic analysis, which is an open problem in machine learning based network traffic analysis. Existing methods often rely on unrealistic assumptions, such as assuming clear-cut distinctions between benign and malicious traffic, or they fall short in capturing the intricate structure and semantics inherent in network traffic data.
> > >
> > > In contrast, FlowRefiner identifies noisy labels based on traffic semantics, refines them through confidence-aware prediction, and enables robust, cross-granularity classification tailored specifically to the unique characteristics of traffic data. We believe our contributions offer a meaningful step forward for machine learning in network traffic analysis and provide a foundation for more resilient real-world applications.
> > >
> > > These innovations align well with the NeurIPS mission to advance machine learning research on challenging and impactful real-world problems, particularly within the *Primary Area: Applications* that we have selected.
> > >
> > > Thank you once again for your thoughtful consideration and feedback.

---

### Official Review · Reviewer_HgNa · 2025-06-22

**Clarity:** 3
**Significance:** 3
**Originality:** 3
**Rating:** 5
**Confidence:** 5

**Summary:**

This paper presents FlowRefiner, a robust and general-purpose framework for network traffic classification that addresses label noise. The framework uses three core components to tackle label noise issues. Experiments on four real-world traffic datasets show that FlowRefiner significantly outperforms existing baselines across different noise ratios.

**Questions:**

1. Have other pre-training methods been considered for semantic extraction? How does it generalize to complex scenarios like encrypted traffic?
2. How are loss functions balanced between flow-level and packet-level tasks? Is there a risk of performance degradation due to task conflicts? Can you provide more details on parameter sharing strategies?
3. Which network devices can FlowRefiner be deployed on? Can existing network devices’ computing resources support deployment, given that traffic classification often requires line-rate processing in many scenarios?
4. Experiments assume uniform noise distribution, but real networks may have localized high noise. How robust is FlowRefiner against non-uniform noise distributions?

**Ethical Concerns:**

["NO or VERY MINOR ethics concerns only"]

**Final Justification:**

Given authors' comprehensive response, I think this work still has some significance, so I raise my score to 5.

**Limitations:**

1. Performance of the noise detection module is constrained by the pre-trained encoder’s semantic extraction capability. If pre-training data differs significantly from the target domain, re-pretraining is needed.
2. FlowRefiner requires multiple data traversals and model updates, making edge device deployment difficult.
3. It primarily targets random label noise. While its effect on class-dependent noise is evaluated, robustness in extreme class-dependent scenarios needs enhancement.

If the authors can solve the above problems, I would be happy to raise the score.

**Quality:**

3

**Strengths And Weaknesses:**

Strengths:

1. The paper is well-structured with precise and professional writing.
2. It introduces a label noise handling framework for general traffic classification, breaking through the limitations of existing methods that focus on specific scenarios like malicious traffic.
3. It uses four datasets from different scenarios, covering noise ratios from 5% to 60%, and compares against comprehensive baselines including traditional methods, pre-trained models, and label noise-specific approaches.

Weaknesses:

1. The pre-trained encoders rely on existing techniques, with innovation focused on framework design rather than underlying representation learning.
2. Key parameters like clustering granularity and majority label counts require manual setting, with no mention of adaptive adjustment mechanisms, which may limit generalization to unknown datasets.
3. Random sampling in packet-level classification may lose critical packet sequence information, and the framework’s adaptability to time-sensitive traffic is not discussed.
4. Computational overhead is not reported, and robustness in real-world scenarios needs further validation.

---

> ### Author Rebuttal · Authors · 2025-07-31
>
> We appreciate the thoughtful review and the constructive feedback! We hope the following clarifications and experiments address your concerns.
>
> **About Pre-trained Encoder.**
>
> > W1: The pre-trained encoders rely on existing techniques, with innovation focused on framework design rather than underlying representation learning.
>
> We clarify that the pre-trained encoder extracts the semantics of the traffic from the raw data for FlowRefiner, which is first proposed as the unique insight for effectively detecting outlier labels as noise.
>
> Besides, the point of our innovation is not in traffic representation learning. As stated in the introduction, our core contribution is the first robust framework to address the critical label noise issue in general traffic classification, a problem that is not handled by previous works.
>
>
> **About Parameters.**
>
> > W2: Key parameters like clustering granularity and majority label counts require manual setting, with no mention of adaptive adjustment mechanisms, which may limit generalization to unknown datasets.
>
>
> As stated in Appendix F, we use the same parameter settings across all datasets, demonstrating that our method achieves strong performance without extensive tuning.
> In addition, our parameter design incorporates moderate adaptivity.
> Specifically, the clustering granularity is used to scale the number of dataset classes, and together they determine the final cluster count.
> Thus, the clustering can better adapt to different datasets, reducing the need for manual adjustment.
>
> Moreover, we are happy to further explore more automated hyperparameter tuning methods in future work.
>
> **About Random Sampling.**
>
> > W3: Random sampling in packet-level classification may lose critical packet sequence information
>
> We utilize random sampling to obtain varied packet samples from the same flow during training, reducing model overfitting to single samples under label noise.
> On the other hand, the critical packet sequence information is captured by the flow-level task with the same share-weighted classifier.
> This design enables our model to leverage both diverse local packet information and global flow-level context, achieving robust performance under label noise.
>
> **Time-sensitive Adaptability.**
>
> > W3: the framework’s adaptability to time-sensitive traffic is not discussed.
>
>
> Our two-level robust classifier addresses this need, providing both flow-level and packet-level heads and thus supporting different deployment requirements (e.g., time-sensitive traffic).
> In time-sensitive scenarios, the packet-level classification could be performed to avoid waiting for entire flows.
>
> The packet classification ability is detailed in Appendix G. Under a 20% noise ratio, FlowRefiner achieves F1 scores above 80% on all datasets, while other packet-level baselines struggle to reach 60%. This demonstrates that our method can perform accurate and robust packet-level classification, making it suitable for time-sensitive traffic scenarios.
>
>
> **Computational Overhead.**
>
> > W4: Computational overhead is not reported.
>
> We reported the time and computational overhead in Limitations of Section 4.5. As stated, our framework currently takes 3 minutes on an RTX 3090 GPU with 2.79 GB of memory.
>
>
> > L2: FlowRefiner requires multiple data traversals and model updates, making edge device deployment difficult.
>
> Note that 'multiple data traversals and model updates' are performed during the training stage.
> In the testing (deployment) stage, FlowRefiner operates identically to existing deep learning traffic methods, introducing no additional procedures or overhead at inference time.
>
>
> **Response to Questions.**
>
> > **Q1**: Have other pre-training methods been considered for semantic extraction? How does it generalize to complex scenarios like encrypted traffic?
>
> We further considered ET-BERT, an encrypted traffic pre-training method, as the encoder of our framework.
> And the evaluation on two classic encrypted traffic datasets is as follows:
>
> | Dataset   | Method       | 20%       | 40%       | 60%       |
> | --------- | ------------ | --------- | --------- | --------- |
> | ISCXVPN   | ET-BERT      | 79.07     | 61.78     | 36.21     |
> | ISCXVPN   | ET-BERT+Ours | **84.69** | **75.82** | **60.58** |
> | CrossPlat | ET-BERT      | 94.49     | 79.30     | 59.40     |
> | CrossPlat | ET-BERT+Ours | **98.68** | **94.15** | **71.87** |
>
> These results demonstrate that our framework consistently improves robustness over the base ET-BERT, especially under high noise ratios, and generalizes well to encrypted traffic scenarios.
>
>
> > **Q2.1**: How are loss functions balanced between flow-level and packet-level tasks?
>
> We do not introduce additional hyperparameters to balance the losses of the flow-level and packet-level tasks.
> Concretely, in each training epoch, we first perform flow-level classification and backpropagate its loss, and then conduct the packet-level task using the same encoder weights.
> This alternating process avoids manual loss weighting and lets the shared encoder learn features beneficial to both tasks.
>
> > **Q2.2**: Is there a risk of performance degradation due to task conflicts?
>
> We are happy to clarify that flow- and packet-level tasks complement each other, rather than causing conflicts.
> As demonstrated in Table 4 of Section 4.4, we remove the packet-level classification branch (i.e., w/o cross-granularity robust classifier), which leads to lower performance across all datasets and noise ratios.
> This result highlights that the joint training of flow-level and packet-level tasks is mutually beneficial, without risk of performance degradation.
>
> > **Q2.3**: Can you provide more details on parameter sharing strategies?
>
> We detailed the parameter sharing strategies in Appendix C, Figure 7.
> In the flow-level classifier, multiple parallel packet encoders process each individual packet from the flow; these encoders share the same weights. The encoded packet features are then aggregated to form the flow representation.
> This design enables the same parameter encoder could also handle packet data and perform packet-level classification.
>
> > **Q3**: Which network devices can FlowRefiner be deployed on? Can existing network devices’ computing resources support deployment, given that traffic classification often requires line-rate processing in many scenarios?
>
>
>
>
>
> To further address efficiency and deployment concerns, we performed additional experiments under a variety of hardware settings, including GPU, CPU, and an edge-device simulation:
>
>
> |                | GPU           | CPU          | Edge Device |
> | -------------- | ----------------- | ---------------- | -------------------------- |
> | Inference Time | 0.24 ms          | 5.31 ms        | 145.07 ms                 |
> | Throughput     | 4226.88 samples/s | 188.26 samples/s | 6.89 samples/s             |
>
> The edge device is simulated via a single-core CPU, 50% CPU quota, 1GB memory with swap disabled, and instruction set limited to AVX2.
> These results show that FlowRefiner can be flexibly deployed on a range of network devices, from high-performance GPUs to constrained edge devices.
>
>
> > **Q4**: Experiments assume uniform noise distribution, but real networks may have localized high noise. How robust is FlowRefiner against non-uniform noise distributions?
>
> We introduce non-uniform (class-dependent) noise evaluation in lines 299-308 of Section 4.2.
> The details are shown in Appendix H, Table 6, Table 7.
> Results show that our method can achieve significant performance advantages over the baselines against non-uniform noise distributions.
>
> Additionally, we also evaluate FlowRefiner under real-world non-uniform noise of CICIDS2017 in line 324-331, Section 4.3.
> The results demonstrate the ability of our method to handle real-world non-uniform noise.
>
> **About Limitations.**
>
> We appreciate your valuable suggestions.
> Future work will explore domain-adaptive pre-training techniques and more extreme class-dependent noise scenarios to enhance robustness further.

---

> > ### Comment · Reviewer_HgNa · 2025-08-04
> >
> > Thank you for your rebuttal. Although the authors have addressed most of my concerns, I still have questions regarding the inference time reported in your response to Q3. While a inference time of 0.24 ms appears quite short, it remains potentially prohibitive for high performance environments such as modern data center networks and high-performance computing (HPC) systems operating at rates of 400 Gbps and above. In such scenarios, the volume of traffic can be so high that packets may be transmitted before classification results are available. Furthermore, current data center switches typically lack dedicated computation resources such as specialized accelerator cards, which further constrains the practical deployment of your solution in these environments.

---

> > > ### Author Response · Authors · 2025-08-05
> > >
> > > We sincerely appreciate your time and constructive feedback. We are happy that most concerns you raised have been addressed.
> > >
> > > We clarify that FlowRefiner achieves inference times that are competitive with state-of-the-art machine learning-based traffic classification methods, and is faster than other Transformer-based baselines. Specifically, FlowRefiner operates at 0.24 ms per sample, compared to 0.36 ms for YaTC and 1.44 ms for ET-BERT.
> > >
> > > We fully acknowledge that efficiency is critical for traffic analysis systems. However, achieving ultra-low latency at extremely high throughputs (e.g., 400 Gbps and beyond) remains an open challenge across all existing machine learning-based traffic analysis methods, and is not a limitation unique to our approach.
> > >
> > > Our core motivation and contribution lie in tackling a longstanding and critical challenge in machine-learning-based network traffic analysis, i.e., ensuring robustness against label noise in general traffic classification. While our current focus centers on this fundamental issue, we fully recognize the importance of system-level optimizations, such as model compression, pruning, and deployment on efficient hardware, for practical deployment. In future work, we plan to explore lightweight architectures and acceleration techniques to further reduce the latency of machine learning-based traffic classification. Moreover, FlowRefiner stands to benefit directly from ongoing advancements in lightweight inference and dedicated hardware platforms as they become more accessible.
> > >
> > > We sincerely appreciate your thoughtful comments, which have significantly contributed to strengthening our work. We will incorporate additional discussions on efficiency in the revised version. Should you have any further suggestions or feedback, we would be grateful to receive them.

---

> > > > ### Comment · Reviewer_HgNa · 2025-08-07
> > > >
> > > > Given your comprehensive response, I think this work still has some significance, so I will raise my score.

---

> > > > > ### Author Response · Authors · 2025-08-07
> > > > >
> > > > > We sincerely appreciate your thoughtful feedback and your willingness to raise the final score. We truly value your recognition and encouragement.

---

> > > > > ### Author Response · Authors · 2025-08-09
> > > > >
> > > > > Thanks for the response and your decision to increase the score. We are glad to hear that the rebuttal addressed your concerns.
> > > > >
> > > > > On a minor note, we wanted to bring to your attention that the score we see is still the same as the score before rebuttal. Though this could be just an issue with OpenReview. Thanks!

---

### Official Review · Reviewer_MEnh · 2025-07-01

**Clarity:** 4
**Significance:** 3
**Originality:** 3
**Rating:** 5
**Confidence:** 5

**Summary:**

The authors proposed FLOWREFINER framework to solve noise-label related network classification problems.
With a three-step workflow, FLOWREFINER can detect outlier labels by clustering, correct noise based on predictor confidence and implement final classification with a cross-granularity classifier.
In experiments, authors compared FLOWREFINER with different network classification methods and label noise learning methods under manully noise added datasets. Results show FLOWREFINER outperform other methods among all noise related scenarios.
Finally authors discussed several aspects that may influence the noise learning performances and future plan of their framework.

**Questions:**

1. Cross-platform dataset consists servral regions and two different operation systems, which make it naturally suitable for better noise setting (for example, India dataset has systematic noise from the perspective of China region data). Can such data setting be used in experiments to furtherly demonstrate the novelty of FLOWREFINER?
2. Authors introduced the influence of different settings for cluster number parameter n, majority label count m. However, it will be better to demonstrate the selection basis of  noise correlation confidences.
3. Network traffic can be processed as sequence or images. Author considerd the traffic as images for MAE pretraining, will there be any difference when we consider such data as sequences like what happened in ET-BERT?

**Ethical Concerns:**

["NO or VERY MINOR ethics concerns only"]

**Final Justification:**

During the rebuttal period, authors provides additional experimental results, especially the one regarding distribution shift, which addresses my concerns. I have updated my score.

**Limitations:**

yes

**Paper Formatting Concerns:**

There is no paper formatting concerns in this paper

**Quality:**

3

**Strengths And Weaknesses:**

Strengths:
1. The motivation and design process of the FLOWREFINER framework is clear and consistent.
2. Detailed designs for each component is practical for solving targted tasks and related experiments are abundant.
3. The writing is clear and easy to ubderstand.

Weakness:
1. The cluster number parameter n, majority label count m and confidence thresholds rely on actual experiment datasets, which might affect the generalization among different scenarios.
2. All datasets' noise come from manually twisting instead of  naturally deterioration by time or locations, which makes the noise easier to detect.

---

> ### Author Rebuttal · Authors · 2025-07-31
>
> We sincerely appreciate your time and the positive feedback!
> We hope our responses below further clarify our work and address your concerns.
>
>
>
>
> **About Hyperparameters.**
> > W1: ... thresholds rely on actual experiment datasets, which might affect the generalization among different scenarios.
>
> We are happy to clarify that we apply the same hyperparameter settings across all datasets in evaluation. The consistent robustness demonstrates that our method achieves strong performance without extensive tuning, highlighting its generalizability among different scenarios.
>
> In addition, our hyperparameter design also incorporates moderate adaptivity.
> Specifically, the clustering granularity parameter is used to scale with the number of classes in each dataset, and together they determine the final cluster count.
> This allows the clustering process to better adapt to varying dataset characteristics, further reducing the need for manual adjustment.
>
>
> **About Manual Noise.**
>
> > W2: All datasets' noise come from manually twisting instead of naturally deterioration by time or locations, which makes the noise easier to detect.
>
> We follow widely-adopted practices of symmetric manual noise in label-noise literature in our evaluation.
> The results show that current methods of traffic classification or label noise learning can hardly achieve good performance under such settings.
>
> Additionally, we evaluated our method on the real-world noisy dataset CICIDS2017 in Sec 4.3, which naturally contains noisy labels according to previous studies [18,19].
> As shown in Figure 3 (c), our method successfully identifies 84.61% of the noisy flows under real-world noise, and the accuracy of our noise detection is 81.92%.
> The results demonstrate the effectiveness of our method under real-world noise.
>
>
> **Response to Questions.**
>
> > **Q1**: "Cross-platform dataset consists servral regions and two different operation systems, which make it naturally suitable for better noise setting (for example, India dataset has systematic noise from the perspective of China region data). Can such data setting be used in experiments to furtherly demonstrate the novelty of FLOWREFINER?"
>
> Thank you for the insightful suggestion. However, the described cross-platform dataset scenario primarily reflects domain shifts rather than actual label noise, i.e., mislabeling within the same categories.
> Since the India and China datasets have different application categories, which are collected according to regional popularity, they cannot directly serve to evaluate under label noise scenarios. We will release our codes and further evaluate FlowRefiner on more scenarios in future work.
>
>
>
>
> > **Q2**: Authors introduced the influence of different settings for cluster number parameter n, majority label count m. However, it will be better to demonstrate the selection basis of noise correlation confidences.
>
> We are happy to provide the detailed discussions about confidence thresholds. The influence to F1 scores of these thresholds on ISCXVPN dataset under different noise ratios is shown as follows.
>
>
> | High-confidence Thresholds | 20%       | 40%       | 60%       |
> | :------------------------- | :-------- | :-------- | :-------- |
> | 0.95                       | 88.25     | 81.32     | 69.92     |
> | 0.90                       | **90.15** | **84.19** | **72.90** |
> | 0.80                       | 86.94     | 81.78     | 72.49     |
> | 0.70                       | 84.03     | 80.68     | 70.44     |
>
>
>
> | Low-confidence Thresholds | 20%       | 40%       | 60%       |
> | ------------------------- | :-------- | :-------- | :-------- |
> | 0.80                      | 87.03     | 83.29     | 70.63     |
> | 0.70                      | **90.15** | **84.19** | **72.90** |
> | 0.60                      | 86.42     | 83.54     | 72.68     |
> | 0.50                      | 84.52     | 82.80     | 72.84     |
>
>
>
>
>
> - The optimal setting for the high confidence threshold is 0.90, which yields the best performance across all noise ratios. When the threshold is set too high (i.e., 0.95), the model becomes overly cautious, failing to relabel flows that could have been corrected, leading to a decline in performance. On the other hand, setting the threshold too low (i.e., 0.70-0.80) can lead to overcorrection by relabeling flows that are uncertain, thereby introducing more potential noise into the raw flows, especially under the high noise ratio.
> - For the low confidence threshold, the optimal setting is 0.70. This setting ensures that the flows outside the clean set’s distribution are reintroduced with their original labels, preserving the diversity of the dataset without overwhelming it with potential noise.
>
>
>
>
> > **Q3**: Network traffic can be processed as sequence or images. Author considerd the traffic as images for MAE pretraining, will there be any difference when we consider such data as sequences like what happened in ET-BERT?
>
>
> We process the traffic into sequences and replace the encoder of FlowRefiner with ET-BERT. The results are shown below:
>
>
>
> | Dataset   | Method       | 20%       | 40%       | 60%       |
> | --------- | ------------ | --------- | --------- | --------- |
> | ISCXVPN   | ET-BERT      | 79.07     | 61.78     | 36.21     |
> | ISCXVPN   | ET-BERT+Ours | **84.69** | **75.82** | **60.58** |
> | CrossPlat | ET-BERT      | 94.49     | 79.30     | 59.40     |
> | CrossPlat | ET-BERT+Ours | **98.68** | **94.15** | **71.87** |
> | USTC      | ET-BERT      | 92.34     | 81.69     | 58.09     |
> | USTC      | ET-BERT+Ours | **95.86** | **91.42** | **79.93** |
> | Malware   | ET-BERT      | 78.03     | 61.49     | 43.38     |
> | Malware   | ET-BERT+Ours | **78.08** | **75.44** | **65.33** |
>
> These results indicate that FlowRefiner maintains strong robustness and significantly outperforms original ET-BERT across datasets and noise levels. Since sequence-based language models typically require significantly greater computational resources than image-based models, FlowRefiner processes traffic as images to remain more efficient and practical in real-world deployments.

---

> > ### Comment · Reviewer_MEnh · 2025-08-05
> > **clarification over hyerparameters**
> >
> > I appreciate authors for clarifying the impact of hyperparameter selection and it is good to know that the proposed method is robust against hyperparameter selection. I am still wonder whether the proposed method is robust against distribution shift which is highly possible when the model is indeed applied to the real setting where application softwares could keep updating.
> > Authors will be appreciated if you can share your thought on it .

---

> > > ### Author Response · Authors · 2025-08-07
> > >
> > > Thank you very much for your thoughtful feedback. We are glad to know that our rebuttal has addressed your earlier concerns, and we appreciate your raising the issue of distribution shift in the real-world setting.
> > >
> > > We would like to clarify that FlowRefiner is specifically designed to tackle label noise issues in general traffic classification scenarios, and distribution shift represents a distinct research direction beyond our original scope.
> > >
> > > Nevertheless, inspired by your valuable suggestion, we conducted an additional experiment to explicitly evaluate FlowRefiner’s robustness against distribution shifts. Specifically, we used realistic traffic data collected by [45] at different dates, following:
> > > - In-Distribution setting: Training and test sets are collected from traffic on the same date.
> > > - Distribution Shift setting: Training and test sets are collected from traffic on different dates, separated by 9 days on average.
> > >
> > > For a clear comparative analysis, we employed a baseline classifier with the same Transformer encoder without our FlowRefiner framework. The results are summarized below.
> > >
> > >
> > > | Method      | In-Distribution  | Distribution Shift     |
> > > | ----------- | --------- | --------- |
> > > | Baseline    | 93.05     | 77.54     |
> > > | FlowRefiner | **97.65** | **91.46** |
> > >
> > > It can be seen that although both models experience reduced F1 scores under distribution shift, FlowRefiner’s performance is much more stable, exhibiting a smaller drop (6.19%) compared to the baseline (15.51%). This demonstrates that, while not specifically designed for distribution shift, FlowRefiner exhibits notable robustness under such conditions.
> > >
> > > We appreciate your insightful comments, which have inspired this additional valuable analysis. We sincerely hope these additional efforts and findings could address your further concerns, and would be grateful if they could be positively considered in your final rating.

---

> > > > ### Comment · Reviewer_MEnh · 2025-08-08
> > > > **it addresses my concerns**
> > > >
> > > > Thanks for your continous efforts on the extra experiments, which address my concern. I have no further comment, thanks.

---

> > > > > ### Author Response · Authors · 2025-08-09
> > > > >
> > > > > Thanks for your constructive feedback. We are glad that our rebuttal has addressed all your concerns. We sincerely thank you for the time and effort spent on reviewing our paper and response.

---

> ### Comment · Area_Chair_BXBk · 2025-08-05
>
> Dear Reviewer MEnh,
>
> Please help go through the rebuttal and participate in discussions with authors. Thank you!
>
> Best regards,
> AC

---

### Official Review · Reviewer_72E3 · 2025-07-03

**Clarity:** 4
**Significance:** 2
**Originality:** 3
**Rating:** 4
**Confidence:** 4

**Summary:**

The paper presents FlowRefiner, a novel framework for traffic classification in the presence of label noise. It introduces a three modular comprising a noise detector, a label corrector, and a cross-granularity classifier. The proposed method is thoroughly evaluated on four datasets under varying levels of label noise, consistently outperforming state-of-the-art baselines.

**Questions:**

1.	What exactly does a dataset sample refer to a flow or packet?

2.	How is label noise introduced in the experiments? Is it added by randomly selecting samples or based on a specific pattern?

3.	Can the authors provide a more detailed qualitative analysis of failure cases, such as confusion between specific classes?

4.	How well does it generalize across different domains? Does the encoder require dataset-specific fine-tuning or post-tuning, especially as traffic characteristics evolve over time?

5.	How do the authors justify the scalability of the framework to real-world deployment?

6.	When stating that "the framework takes 3 minutes on a GPU," does this refer to training time, inference time, or the full pipeline duration?

**Ethical Concerns:**

["NO or VERY MINOR ethics concerns only"]

**Limitations:**

The authors are upfront about the framework’s computational requirements, which may pose scalability challenges as mentioned above. I suggest providing a more detailed analysis, including separate reporting of training and inference times. For future work, the authors could explore various acceleration techniques such as model distillation and pruning to improve efficiency.

**Quality:**

3

**Strengths And Weaknesses:**

Strengths:
1.	The paper presents a well-designed framework for traffic classification under noisy labels, addressing an important and impactful problem in the community.
2.	The design of framework is well-motivated and supported by comprehensive experiments.
3.	The paper is clearly written and well-organized, making it easy for readers to follow.



Weaknesses:
Regarding experiment setups and result analysis:
1.	The definition of a dataset sample is unclear — does it refer to flows or packets?
2.	Additionally, the process for introducing label noise is not well explained. Is the noise added by randomly selecting samples or by following a specific pattern?
3.	While real-world label correction accuracy is reported, a more detailed qualitative analysis of failure cases (e.g., confusion between specific classes) would help strengthen the claims.

Regarding generalizability:
1. The performance of the framework appears to rely heavily on the pre-trained encoder, whose generalizability across different domains remains unclear. It is also not specified whether the encoder requires dataset-specific fine-tuning or post-tuning, especially as traffic characteristics may evolve over time.

Regarding scalability:
4.	The dataset scale appears limited, with only a few thousand samples per dataset. The authors should clarify how such experiments support claims of real-world scalability.
5.	The reported runtime—"the framework takes 3 minutes on a GPU"—is ambiguous. It is unclear whether this refers to training time, inference time, or the entire pipeline.

---

> ### Author Rebuttal · Authors · 2025-07-31
>
> Thank you for your time and careful review of our work!
> We hope our responses below address your concerns.
>
> **Sample Definition.**
> > **Q1**: The definition of a dataset sample is unclear — does it refer to flows or packets?
>
> Each dataset sample in our experiments corresponds to a network flow, which is defined by the standard 5-tuple (source IP, destination IP, source port, destination port, and protocol). This is consistent with common practice in network traffic classification literature (e.g., [30, 33]).
>
>
> Additionally, in our Cross-Granularity Robust Classifier, each packet is sampled from an individual corresponding flow sample to perform pcaket-level classification task, which is detailed in Section 3.3.
> We will clarify these definitions in the revision for better reproducibility.
>
>
>
>
> **Noise Introducing Process.**
>
> > **Q2**: How is label noise introduced in the experiments? Is it added by randomly selecting samples or based on a specific pattern?
>
> We introduce label noise following established protocols in label noise learning, e.g., [22][23][24] in our manuscript.
> Our experiments involve symmetric noise, which is added by randomly selecting a specified percentage of samples (e.g., 20%, 40%, 60%) and uniformly flipping their labels to other classes.
>
> Furthermore, we consider class-dependent noise based on traffic similarity patterns for evaluation in Section 4.2 and Appendix H. As detailed in Appendix H, we introduce class-dependent scenarios by assigning higher mislabeling probabilities between semantically similar classes of traffic.
>
>
>
>
>
>
>
> **Failure Cases.**
>
> > **Q3**: Can the authors provide a more detailed qualitative analysis of failure cases, such as confusion between specific classes?
>
>
> Thank you for your suggestion. We present a more detailed qualitative analysis of failure cases using the confusion matrices of ISCXVPN under different noise rates.
>
> At a 20% noise rate, the most significant off-diagonal errors are observed between CHAT and VOIP, FTP and VOIP, and BROWSING and Streaming. These errors typically occur between semantically similar classes. For instance, both CHAT and VOIP are real-time communication applications with overlapping traffic patterns, making them challenging to distinguish, especially under label noise.
>
> As the noise rate increases to 40% and 60%, we observe an increase in confusion among these related classes, particularly among CHAT, VOIP, and FTP. Despite the higher label noise, the main diagonal entries remain dominant, demonstrating the robustness of our method against label noise.
>
> We will include the confusion matrix in the revision.
>
>
>
> **About Generalizability.**
> > **Q4**: How well does it generalize across different domains? Does the encoder require dataset-specific fine-tuning or post-tuning, especially as traffic characteristics evolve over time?
>
> In our framework, the pre-trained encoder used in the noise detector is frozen and does not require any dataset-specific fine-tuning or post-tuning.
> As stated and proved in recent advances (e.g., Ref [30, 32, 33]), well-designed pre-trained encoders can generalize across diverse traffic domains and downstream tasks.
> Our results on multiple datasets, including the real-world CICIDS dataset, further support this generalizability.
>
> We agree that as network traffic characteristics evolve, continual adaptation is an important direction, and we plan to explore online adaptation and domain adaptation techniques in future work.
>
>
>
>
>
> **About Scalability.**
>
> > **Q5**: How do the authors justify the scalability of the framework to real-world deployment?
>
> Our evaluation follows the classic datasets commonly used in the field of traffic analysis [30][32][33], which contain only thousands of flows.
> However, to demonstrate real-world robustness and scalability, we conduct experiments (Section 4.3) on the CICIDS2017 dataset, which contains over 200,000 flows.
> Our framework maintains robust performance on this large-scale dataset, indicating its practical applicability for real-world deployment. We will highlight these large-scale results more clearly in the revised manuscript.
>
>
>
>
> > **Q6**: When stating that "the framework takes 3 minutes on a GPU," does this refer to training time, inference time, or the full pipeline duration
>
> The “3 minutes on a GPU” refers to the average total runtime (176.28 seconds) of the full pipeline (including noise detector, label correction, and robust classification) across all four benchmark datasets, measured on a single NVIDIA RTX 3090 GPU.
> For clarity, we also provide the detailed runtime for each component on all datasets in the table below.
>
> | Time (s)              | ISCXVPN  | CrossPlat | USTC     | Malware  |
> | --------------------- | -------- | --------- | -------- | -------- |
> | Noise Detector        | 7.2895   | 16.9373   | 6.4972   | 8.2633   |
> | Label Corection       | 39.6972  | 63.5462   | 34.8589  | 45.1194  |
> | Robust Classification | 99.7816  | 175.7102  | 90.3483  | 117.0476 |
> | Total Pipeline        | 146.7719 | 256.1984  | 131.7097 | 170.4369 |

---

> ### Comment · Area_Chair_BXBk · 2025-08-05
>
> Dear Reviewer 72E3 ,
>
> Please help go through the rebuttal and participate in discussions with authors. Thank you!
>
> Best regards,
> AC

---

### Decision · Program_Chairs · 2025-09-17

**Decision:**

Accept (poster)

**Comment:**

This paper introduces FlowRefiner, a modular framework for traffic classification under noisy labels, consisting of a noise detector, a label corrector, and a cross-granularity classifier. The strengths of the work lie in its clear problem motivation, well-structured design, strong experimental validation across multiple datasets and noise ratios, and clarity of presentation. The main weaknesses are related to experimental evaluation, such as questions around noise modeling, scalability, parameter sensitivity, and robustness to more realistic deployment settings. During the rebuttal, the authors provided additional experiments and clarifications, which addressed many concerns on experimental rigor and robustness. As all the reviewers are positive on this paper after the rebuttal, I would like to recommend acceptance.